# Imagine Beyond! Distributionally Robust Auto-Encoding for State Space Coverage in Online Reinforcement Learning

**Nicolas Castanet**
Sorbonne Université, CNRS, ISIR, F-75005 Paris, France
`nicolas.castanet@isir.upmc.fr`

**Olivier Sigaud**
Sorbonne Université, CNRS, ISIR, F-75005 Paris, France
`olivier.sigaud@isir.upmc.fr`

**Sylvain Lamprier**
Univ Angers, LERIA, Angers, France
`sylvain.lamprier@univ-angers.fr`

## Abstract

Goal-Conditioned Reinforcement Learning (GCRL) enables agents to autonomously acquire diverse behaviors, but faces major challenges in visual environments due to high-dimensional, semantically sparse observations. In the online setting, where agents learn representations while exploring, the latent space evolves with the agent's policy, to capture newly discovered areas of the environment. However, without incentivization to maximize state coverage in the representation, classical approaches based on auto-encoders may converge to latent spaces that over-represent a restricted set of states frequently visited by the agent. This is exacerbated in an intrinsic motivation setting, where the agent uses the distribution encoded in the latent space to sample the goals it learns to master. To address this issue, we propose to progressively enforce distribuional shifts towards a uniform distribution over the full state space, to ensure a full coverage of skills that can be learned in the environment. We introduce DRAG (Distributionally Robust Auto-Encoding for GCRL), a method that combines the $\beta$-VAE framework with Distributionally Robust Optimization. DRAG leverages an adversarial neural weighter of training states of the VAE, to account for the mismatch between the current data distribution and unseen parts of the environment. This allows the agent to construct semantically meaningful latent spaces beyond its immediate experience. Our approach improves state space coverage and downstream control performance on hard exploration environments such as mazes and robotic control involving walls to bypass, without pre-training nor prior environment knowledge.

## 1 Introduction

Goal-Conditioned Reinforcement Learning (GCRL) enables agents to master diverse behaviors in complex environments without requiring predefined reward functions. This capability is particularly valuable for building autonomous systems that can adapt to various tasks, especially in navigation and robotics manipulation environments [Plappert et al., 2018, Rajeswaran et al., 2018, Tassa et al., 2018, Yu et al., 2021]. However, when working with visual inputs, agents face significant challenges:

39th Conference on Neural Information Processing Systems (NeurIPS 2025).

observations are high-dimensional and lack explicit semantic information, making intrinsic goal generation for exploration, reward calculation, and policy learning substantially more difficult.

A common approach to address these challenges involves learning a compact latent representation of the observation space, that captures semantic information while reducing dimensionality [Nair et al., 2018, Colas et al., 2018, Pong et al., 2019, Hafner et al., 2019a, Laskin et al., 2020, Gallouédec and Dellandréa, 2023]. Assuming a compact - information-preserving - representation that encodes the main variation factors from the whole state space, agents can leverage latent codes as lower-dimensional inputs. In the GCRL setting, agents are conditioned with goals encoded as latent codes, usually referred to as skills [Campos et al., 2020], which reduces control noise and enables efficient training. Thus, many works build agents on such pre-trained representations of the environment [Mendonca et al., 2023, Zhou et al., 2025], but usually leave aside the question of the collection of training data, by assuming the availability of a state distribution from which sampling is efficient. Without such knowledge, some methods use auxiliary exploration policies for data collection [Campos et al., 2020, Yarats et al., 2021, Mendonca et al., 2021], such as maximum entropy strategies Hazan et al. [2019] or curiosity-driven exploration Pathak et al. [2017], but these often struggle in high-dimensional or stochastic environments due to density estimation and dynamics learning difficulties.

An alternative, which we focus on in this work, is the *online* setting: the representation is learned jointly with the agent's policy, using rollouts to train an encoder-decoder. This allows the representation to evolve with the agent's progress and potentially cover the full state space. Unlike auxiliary exploration, GCRL-driven representation learning aligns training with a meaningful behavioral distribution, which naturally acts as a curriculum. A representative approach is RIG [Nair et al., 2018], where a VAE encodes visited states, and latent samples from the prior are used as "imagined" goals—creating a feedback loop between representation and policy learning.

However, this process suffers from key limitations. A common critique is that continual encoder training leads to **distributional shift**—a well-known issue in machine learning—which destabilizes policy learning and reduces exploration diversity. In this collaborative setting, we identify two distinct sources of shift: one from the *agent's perspective*, where the meaning of the latent codes it receives as inputs continuously evolves; and one from the *encoder's perspective*, in the distribution of visited states to be encoded during rollouts. While policy instability caused by distributional shift from the agent's perspective can be mitigated using a delayed encoder, we argue that distributional shift in the encoder's input data—i.e., the states reached during rollouts—is not only desirable, but essential for exploring the environment and expanding the representation. Rather than limiting such shift, we propose to anticipate and deliberately steer it using a principled method, ensuring that it benefits exploration and learning rather than undermining them.

Our main contribution is to leverage **Distributionally Robust Optimization (DRO)** [Delage and Ye, 2010] to guide the evolution of the representation. By integrating DRO with a $\beta$-VAE [Higgins et al., 2017a], we introduce DRAG (*Distributionally Robust Auto-Encoding for GCRL*), which uses an adversarial weighter to emphasize underrepresented states. This allows the agent to build latent spaces that generalize beyond its current experience, progressively covering the state space.

Our contributions are:

- We introduce a DRO-based VAE framework tailored to GCRL.

- We reinterpret SKEW-FIT [Pong et al., 2019] as a non-parametric instance of DRO-VAE.

- We propose DRAG, a - more stable - parametric DRO-VAE approach to encourage state coverage through adversarial neural weighting.

- We show that when encoder learning anticipates distributional shift, explicit exploration strategies become unnecessary in RIG-like methods; the latent prior alone generates meaningful goals. This enables focusing on selecting goals of intermediate difficulty (GOIDs Florensa et al. [2018]) to improve sample efficiency.

Our approach improves state space coverage and downstream control performance on hard exploration environments such as mazes and robotic control involving walls to bypass, without pre-training nor prior environment knowledge.

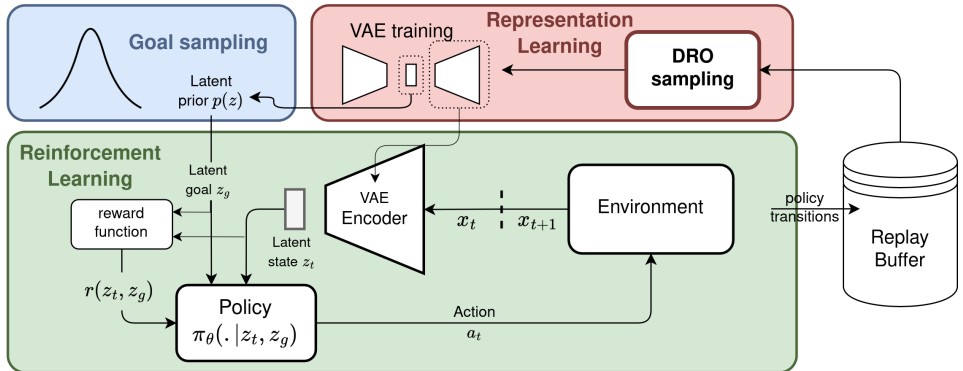

Figure 1: General framework of online VAE representation learning in RL. **Green:** RL loop using the VAE encoder to convert high-dimensional states $s_t$ to latent states $z_t$. **Blue:** latent goal $z_g$ sampling (from prior distribution or replay buffer) and selection. **Red:** Representation Learning with VAE, using data from the replay buffer combined with Distributionally Robust Optimisation (DRO).

## 2 Background & Related Work

### 2.1 Problem Statement: Unsupervised Goal-conditioned Reinforcement Learning

In this work, we consider the multi-goal reinforcement learning (RL) setting, defined as an extended Markov Decision Process (MDP) $\mathcal{M} = <S, T, A, G, R_g, S_0, X>$, where $S$ is a set of continuous states, $T$ is the transition function, $A$ the set of actions, $S_0$ the distribution of starting states, $X$ the observation function and $R_g$ the reward function parametrized by a goal $g \in G$. In our unsupervised setting, we are interested in finding control policies that are able to reach any state in $S$ from the distribution of starting states $S_0$. Thus, we consider that $G \equiv S$. $S$ being continuous, we set the reward function $R_g$ as depending on a threshold distance $\delta$ from the goal $g$: for any state $s \in S$, we consider the sparse reward function $R_g(s) = \mathbb{1}[||s - g||_2 < \delta]$. $R_g(s) > 0$ is only possible once per trajectory. For simplicity, we also consider that any pixel observation $x \in X$ corresponds to a single state $s \in S$. Thus, given an horizon $T$, the optimal policy is defined as $\pi^* = \max_\pi \mathbb{E}_{g \in G} \mathbb{E}_{\tau \sim \pi(\tau|g)} [\sum_{t=0}^{T} R_g(s_t)]$, where $\tau = (s_0, a_1, s_1, ..., s_T)$ is a trajectory, and $\pi(\tau|g)$ is the distribution of trajectories given $g$ in the MDP when following policy $\pi$.

### 2.2 Multi-task Intrinsically Motivated Agents

Multitask Intrinsically Motivated Agents provide a powerful framework in GCRL to tackle unsupervised settings by enabling agents to self-generate and pursue diverse tasks without external prior knowledge of the environment via an *intrinsic goal distribution*. This approach has proven effective for complex problems such as robotic control and navigation, and has also shown benefits in accelerating learning in supervised tasks where goals are known in advance [Colas et al., 2018, Ren et al., 2019, Hartikainen et al., 2020, Gallouédec and Dellandréa, 2023]. Various criteria have been investigated for the formulation of the intrinsic goal distribution. Many of them focus on exploration, to encourage novelty or diversity in the agent's behavior [Warde-Farley et al., 2018, Pong et al., 2019, Pitis et al., 2020, Gallouédec and Dellandréa, 2023, Kim et al., 2023]. Among them, MEGA [Pitis et al., 2020] defines a density estimator $p_t^S$ from the buffer (e.g., via a KDE) and samples goals at the tail of the estimated distribution to foster exploration. SKEW-FIT Pong et al. [2019] maximizes the entropy of the behavior distribution. It performs goal sampling from a skewed distribution $p_{\text{skewed}_t}(\mathbf{s})$, designed as an importance resampling of samples from the buffer with a rate $1/p_t^S$ to simulate sampling from the uniform distribution. Other approaches are focused on control success and agent progress, by looking at goals that mostly benefit improvements of the trained policy. This includes learning progress criteria [Colas et al., 2019], or the selection of goals of intermediate difficulty (GOIDs) [Sukhbaatar et al., 2017, Florensa et al., 2018, Zhang et al., 2020, Castanet et al., 2022], not too easy or too hard to master for the agent, depending on its current level. Approaches from that family, such as GOALGAN [Florensa et al., 2018] or SVGG [Castanet et al., 2022] usually rely on an auxiliary network that produces a GOIDs distribution based on a success predictor.

## 2.3 Online Representation Learning with GCRL

Variational Auto Encoders (VAEs) present appealing properties when it comes to learning latent state representations in RL. With their probabilistic formulation, the observation space can be represented by the latent prior distribution, which enables several operations to take place, such as goal sampling in GCRL [Nair et al., 2018, Pong et al., 2019, Gallouédec and Dellandréa, 2023] and having access to the log-likelihood of trajectories for model-based RL and planning [Higgins et al., 2017b, Hafner et al., 2019b,a, Lee et al., 2020b]. The seminal work RIG [Nair et al., 2018], which is the foundation of our paper, is an online GCRL method that jointly trains a latent representation and a policy $\pi(a \mid z_x, z_g)$, where $z_g$ is a goal sampled in the latent space, and $z_x = q_\psi(x)$ is a VAE encoding of observation $x$ of the current state. During training, the agent samples a goal $z_g \sim p(z)$, with $p(z)$ the prior (typically $\mathcal{N}(0, I)$), and performs a policy rollout during $T$ steps or until the latent goal and the encoded current state are close enough. The policy is then optimized via policy gradient, using e.g. a sparse reward in the latent space, and the visited states are inserted in a training buffer for the VAE. This framework enables the agent to autonomously acquire diverse behaviors without extrinsic rewards, by aligning representation learning and control. The policy collects new examples for the VAE training, which in turn produces new goals to guide the policy, implicitly defining an automatic exploration curriculum. To avoid exploration bottlenecks, which is the main drawback of RIG, the SKEW-FIT principle introduced in the previous section for the sampling of uniform training goals was also applied in the context of GCRL representation learning, on top of RIG. SKEW-FIT for visual inputs [Pong et al., 2019] is, to our knowledge, the most related approach to our work, which can be seen as an instance of our framework, as we show below.

Beyond generative models based on VAE, other types of encoder-decoder approaches have been introduced in the context of unsupervised RL, including normalizing flows [Lee et al., 2020a] and diffusion models [Emami et al., 2023], each offering different trade-offs in terms of expressivity, stability, and sample quality. In addition, contrastive learning methods [Oord et al., 2018, Srinivas et al., 2020, Stooke et al., 2021, Lu et al., 2019, Li et al., 2021, Aubret et al., 2023] have been employed to learn compact and dynamic-aware representations, without relying on reconstruction-based objectives. Some methods rely on pre-trained generalistic models such as DinoV2 to compute semantically meaningful features from visual observations [Zhou et al., 2025], although usually inducing additional computational cost.

In this work, we use the $\beta$-VAE framework [Higgins et al., 2017a], for simplicity and to follow the main trend initiated by RIG Nair et al. [2018]. However, the principle introduced in Section 2.4 could easily be applied to many other representation learning frameworks. The general framework of online VAE representation learning in GCRL is depicted in Figure 1. Compared to RIG, it includes a DRO resampling component, which we discuss in the following.

## 2.4 Distributionally Robust Optimization

This section introduces the general principles of Distributionally Robust Optimization (DRO) [Delage and Ye, 2010, Ben-Tal et al., 2013, Duchi et al., 2021], developed in the context of supervised machine learning to address the problem of distributional shift, which happens when a model is deployed on a data distribution different from the one used for its training. DRO proposes to anticipate possible shifts by optimizing model performance against the worst-case distribution within a specified set around the training distribution. Formally, given a family of possible data distributions $\mathcal{Q}$, DRO considers the following adversarial risk minimization problem: $\min_{\theta \in \Theta} \max_{q \in \mathcal{Q}} \mathbb{E}_{(x,y) \sim q} [\ell(f_\theta(x), y)]$, with $\ell$ a specified loss function which compares the prediction $f_\theta(x)$ with a given ground truth $y$.

In the absence of a predefined uncertainty set $\mathcal{Q}$, DRO methods strive to define such an uncertainty set relying on heuristics. This has been the subject of many research papers, see [Rahimian and Mehrotra, 2019] for a broad and comprehensive review of these approaches. In the following, we build on the formulation proposed in [Michel et al., 2022], which considers $\mathcal{Q}$ as the set of distributions whose KL-divergence w.r.t. the training data distribution $p$ is upper-bounded by a given threshold $\delta$.

**Likelihood Ratios Reformulation** Assuming $\mathcal{Q}$ as a set of distributions that are absolutely continuous with respect to $p$[1], the inner maximization problem of DRO can be reformulated using importance weights $r(x, y)$ such that $q = rp$ [Michel et al., 2022]. In that case, we have:

$\mathbb{E}_{(x,y)\sim q} [\ell(f_\theta(x), y)] = \mathbb{E}_{(x,y)\sim p} [r(x, y)\ell(f_\theta(x), y)]$, which is convenient as training data is assumed to follow $p$.

Given a training dataset $\Gamma = \{(x_i, y_i)\}_{i=1}^N$ sampled from $p$, the optimization problem considered in [Michel et al., 2022] is then defined as:

$$\min_\theta \max_r \frac{1}{N} \sum_{i=1}^N r(x_i, y_i)(\ell(f_\theta(x_i), y_i) - \lambda \log r(x_i, y_i)) \qquad \text{s.t.} \quad \frac{1}{N} \sum_{i=1}^N r(x_i, y_i) = 1, \quad (1)$$

where the constraint ensures that the $q$ function keeps a valid integration property for a distribution (i.e., $\int_{\mathcal{X}, \mathcal{Y}} q(x, y)dxdy = 1$). The term $\lambda \log r$ is a relaxation of a KL constraint, which ensures that $q$ does not diverge too far from $p$[2]. $\lambda$ is an hyper-parameter that acts as a regularizer ensuring a trade-off between generalization to shifts (low $\lambda$) and accuracy on training distribution (high $\lambda$).

From this formulation, we can see that the risk associated to a shift of test distribution can be mitigated simply by associating adversarial weights $r_i := r(x_i, y_i)$ to every sample $(x_i, y_i)$ from the training dataset, respecting $\bar{r} := \frac{1}{N} \sum_{i=1}^N r_i = 1$. That said, $r$ can be viewed as proportional to a categorical distribution defined on the components of the training set.

**Analytical solution:** Given any function $h : \mathcal{X} \to \mathbb{R}$, the distribution $q$ that maximizes $\mathbb{E}_q[h(x)] + \lambda \mathcal{H}_q$, with $\mathcal{H}_q$ the Shannon entropy of $q$, is the maximum entropy distribution $q(x) \propto e^{h(x)/\lambda}$. Thus, we can easily deduce that the inner maximization problem of (1) has an analytical solution in $r_i = N \dfrac{e^{l(f_\theta(x_i), y_i))/\lambda}}{\sum_{j=1}^N e^{l(f_\theta(x_j), y_j))/\lambda}}$ (proof in Appendix C.1). The spread of $\mathcal{Q}$ is controlled with a temperature weight $\lambda$, which can be seen as the weight of a Shannon entropy regularizer defined on discrepancies of $q$ regarding $p$.

**Solution based on likelihood ratios:** While appealing, it is well-known that the use of this analytical solution for $r$ may induce an unstable optimization process in DRO, as weights may vary abruptly for even very slight variations of the classifier outputs. Moreover, it implies individual weights, only interlinked via the outputs from the classifier, while one could prefer smoother weight allocation regarding inputs. This is particularly true for online processes like our RL setting, with new training samples periodically introduced in the learning buffer.

Following Michel et al. [2022, 2021], we rather focus in our contribution in the next section on likelihood ratios defined as functions $r_\psi(x, y)$ parameterized by a neural network $f_\psi$, where we set:

$$r_\psi(x_i, y_i) = n \frac{\exp^{f_\psi(x_i, y_i)}}{\sum_{j=1}^n \exp^{f_\psi(x_j, y_j)}} \quad , \forall \text{ mini-batch } \{(x_j, y_j)\}_{j=1}^n, \qquad (2)$$

where $f_\psi$ is periodically trained on mini-batches of $n$ samples from the training set, using fixed current $\theta$ parameters, according to the unconstrained inner maximization problem of (1) for a given number of gradient steps. This parameterization enforces the validity constraint at the batch-level, through batch normalization hard-coded in the formulation of $r_\psi$. Though it does not truly respect the full validity constraint from (1) in the case of small batches, this performs well for commonly used batch sizes in many classification benchmarks [Michel et al., 2022]. Classifiers obtained through the alternated min-max optimization of (1) are more robust to distribution shifts than their classical counterparts. Using shallow or regularized networks $f_\psi$ is advised, as strong Lipschitz-ness of $r(x, y)$ allows to treat similar samples similarly in the input space, which guarantees better generalization and stability of the learning process. These generalization and stability properties lack to non-parametric versions of DRO, such as a version using the analytical solution for inner-maximization presented above, which could be viewed as the optimal $r_\psi$ based on an infinite-capacity neural network $f_\psi$. In the next section, we build on this framework to set a representation learning process for RL, that encourages the agent to explore.

---

[1] In the situation where all distributions in $\mathcal{Q}$ are absolutely continuous with respect to $p$, for all measurable subset $A \subset X \times Y$ and all $q \in \mathcal{Q}$, $q(A) > 0$ only if $p(A) > 0$.

[2] This can be seen easily, observing that: $KL(q||p) = \int q(x) \log(q(x)/p(x))dx = \int p(x)r(x) \log r(x)dx$.

# 3 Distributionally Robust Auto-Encoding for GCRL

To anticipate distributional shift naturally arising in GCRL with online representation learning, we first propose the design of a DRO-VAE approach, which was never considered in the literature to the best of our knowledge[3]. Then, we include it in our GCRL framework, named DRAG, see Figure 1.

## 3.1 DRO-VAE

Classic VAE learning aims at minimizing the negative log-likelihood: $\mathcal{L} = -\mathbb{E}_{x \sim p(x)} \log p_{\theta,\phi}(x)$, with $p_{\theta,\phi(x)}$ the predictive posterior, which can be written as: $p_{\theta,\phi}(x) = \int p(z) p_\theta(x|z) dz$, where $p(z)$ is a prior over latent encoding of the data $x$, commonly taken as $\mathcal{N}(0, I)$, and $p_\theta(x|z)$ is the likelihood of $x$ knowing $z$ and the parameters of the decoding model $\theta$. Given that this marginalization can be subject to very high variance, the idea is to use an encoding distribution $q_\phi(z|x)$ to estimate this generation probability [Kingma and Welling, 2013]. For any distribution $q_\phi$ such that $q_\phi(z|x) > 0$ for any $z$ with $p(z) > 0$, we have: $p_{\theta,\phi}(x) = \mathbb{E}_{z \sim q_\phi(z|x)} p(z) p_\theta(x|z) / q_\phi(z|x)$.

In our instance of the DRO framework, we thus consider the following optimization problem:

$$\min_{\theta,\phi} \max_{\xi \in \Xi} -\mathbb{E}_{x \sim \xi(x)} \log p_{\theta,\phi}(x), \tag{3}$$

where $\Xi$ is the uncertainty set of distributions of our DRO-VAE approach. As in standard DRO, we introduce a weighting function $r : \mathcal{X} \to \mathbb{R}^+$ which aims at modeling $\frac{\xi}{p}$ for any distribution $\xi \in \Xi$, and respects both validity (i.e., $\mathbb{E}_p r(x) = 1$) and shape constraints (i.e., $KL(\xi||p) \leq \epsilon$, for a given pre-defined $\epsilon > 0$). Relaxing the KL constraint by introducing a $\lambda$ hyper-parameter, we can get a similar optimization problem as in classical DRO. However, as $\log p_{\theta,\phi}(x)$ is intractable directly, we consider a slightly different objective:

$$\min_{\theta,\phi} -\mathbb{E}_{x \sim p} r^*(x) \log p_{\theta,\phi}(x), \tag{4}$$

$$\text{with} \quad r^* = \arg\max_{r:\mathbb{E}_p r=1} -\mathbb{E}_{x \sim p} r(x) \tilde{\mathcal{L}}_{\theta,\phi}(x) - \lambda \mathbb{E}_{x \sim p} r(x) \log r(x),$$

where the only difference is that the inner maximization considers an approximation $\tilde{\mathcal{L}}_{\theta,\phi}(x) \approx \log p_{\theta,\phi}(x)$. $\tilde{\mathcal{L}}_{\theta,\phi}(x)$ is estimated via Monte-Carlo importance sampling, as $\tilde{\mathcal{L}}_{\theta,\phi}(x) = \log \sum_{j=1}^M \exp(\log p_\theta(x|z^j) + \log p_\theta(z^j) - \log q_\phi(z^j|x)) - \log(M)$ given $M$ samples $z^j$ from $q_\phi(z^j|x)$ for any $x$, which can be computed accurately (without loss of low log values) using the LogSumExp trick.

This formulation suggests a learning algorithm which alternates between updating the weighting function $r$ and optimizing the VAE. At each VAE step, the encoder-decoder networks are optimized considering a weighted version of the classical ELBO. Denoting as $r$ the weighting function adapted for current VAE parameters via (4), we have:

$$\mathbb{E}_{x \sim p(x)} r(x) \log p_{\theta,\phi}(x) \geq \mathbb{E}_{x \sim p(x)} r(x) \mathbb{E}_{z \sim q_\phi(z|x)} \log \frac{p(z) p_\theta(x|z)}{q_\phi(z|x)}$$

$$\approx \sum_{i=1}^n \frac{r(x_i)}{n} \left( \frac{1}{m} \sum_{j=1}^m \log p_\theta(x_i|z_i^j) - KL(q_\phi(z|x_i)||p(z)) \right), \tag{5}$$

where this approximated lower-bound $\mathcal{L}_{\theta,\phi,r}^{\text{DRO-VAE}}(\{x_i\}_{i=1}^n)$ can be estimated at each step via Monte-Carlo based on mini-batches of $n$ data points $(x_i)_{i=1}^n$ from the training buffer and $m$ latent codes $(z_i^j)_{j=1}^m$ for each data point $x_i$. Optimization is performed using the reparameterization trick, where each latent code $z_i^j$ is obtained from a deterministic transformation of a white noise $\epsilon_i^j \sim \mathcal{N}(0, I)$.

## 3.2 DRAG

Plugging our DRO-VAE in our GCRL framework as depicted in Figure 1 thus simply comes down to weight (of resample) each sample $x_i$ taken from the replay buffer with a weight $r_i$.

---

[3]This is not surprising, as in classical VAE settings, the aim is to model $p$ with the highest fidelity.

As shown in Section 2.4, classical DRO maximization in Equation (4) has a closed-form solution: $r_i \propto e^{-\tilde{\mathcal{L}}_{\theta,\phi}(x_i)/\lambda}$. In Appendix C.2, we show that in our GCRL setting, this reduces to the SKEW-FIT method, where VAE training samples are resampled based on their $p_{skewed}$ distribution.

We claim that the instability of non-parametric DRO, well-known in the context of supervised ML, is amplified in our online RL setting, where the sampling distribution $p$ depends on the behavior of a constantly evolving RL agent. Our DRAG method thus considers the parametric version of the weighting function, implying a neural weighter $f_\psi : X \to \mathbb{R}$ as defined in Equation (2), trained periodically for a given number of gradient steps on the inner maximization problem of (4).

In our experiments, we use for our weighter $f_\psi$ a similar CNN architecture as the encoder of the VAE, but with a greatly smaller learning rate for stability (as it induces a regularizing lag behind the encoder, and hence enforces a desirable smooth weighting w.r.t. the input space). We also use a delayed copy of the VAE to avoid instabilities of encoding from the agent's perspective. The full pseudo-code of our approach is given in Algorithm 1 in Appendix B.

## 4 Experiments

Our experiments seek to highlight the impact of DRAG on the efficiency of GCRL from pixel input[4]. As depicted in Figure 2, we structure this section around two experimental steps that seek to answer the two following research questions in isolation:

• **Representation Learning strategy**: Does DRAG helps overcoming exploration bottlenecks of RIG-like approaches? (Figure 2a)

• **Latent goal sampling strategy**: What is the impact of additional intrinsic motivation when using the representation trained with DRAG? (Figure 2b)

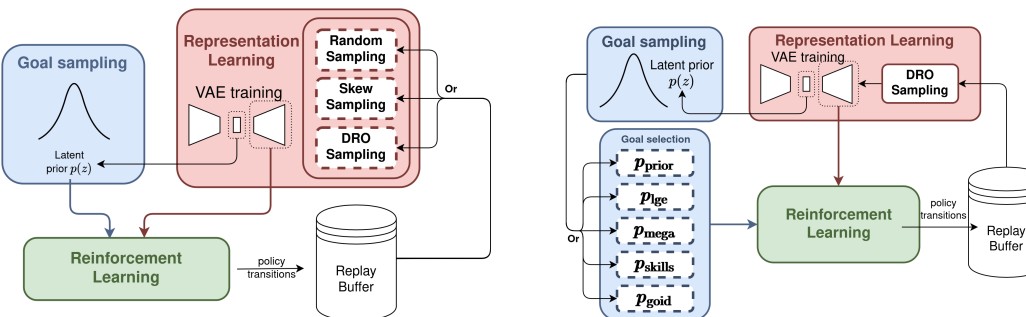

(a) Representation Learning strategy experiments.  (b) Latent Goal selection strategy experiments

Figure 2: Our two questions: (a) how does DRAG perform as a representation learning approach? (b) how does DRAG impact goal sampling approaches from the literature?

In all experiments, the policy $\pi_\theta(\cdot|z, z_g)$ is trained using the TQC off-policy RL algorithm [Kuznetsov et al., 2020], conditioned on the latent state and goal. Learning is guided by a sparse reward in the latent space, defined as $R_g(s) = r(z_x, z_g) = \mathbb{1}\left[|z_x - z_g|_2 < \delta\right]$, where $x$ stands as the pixel mapping of $s$ and $z_x$ its encoding. Experimental details are given in Appendix A.

**Evaluation**  Our main evaluation metric is the *success coverage*, which measures the control of any policy $\pi$ on the entire space of states, defined as:

$$S(\pi) = \frac{1}{|\hat{\mathcal{G}}|} \sum_{g \in \hat{\mathcal{G}}} \mathbb{E}\left[\mathbb{1}[\exists s \in \tau, ||s - g||_2 < \delta|\tau \sim \pi(.|z_0 = q_\psi(X(s_0)), z_g = q_\psi(X(g))), s_0 \sim S_0]\right],$$

where $\hat{\mathcal{G}}$ is a test set of goals evenly spread on $S$, and $X(.)$ stands for the projection of the input state to its pixel representation. Note that goal achievement is measured in the true state space. The knowledge of $S$ is only used for evaluation metrics, remaining hidden to the agent.

---

[4]The code is available at https://github.com/nicolascastanet/DRAG

**Environments** We consider two kinds of environments, with observations as images of size $82 \times 82$. Additional results on image of size $128 \times 128$ are also provided in appendix D.5. In *Pixel continuous PointMazes*, we evaluate the different algorithms over 4 hard-to-explore continuous point mazes. The action space is a continuous vector $(\delta x, \delta y) = [0, 1]^2$. Episodes start at the bottom left corner of the maze. Reaching the farthest area requires at least 40 steps in any maze. States and goals are pixel top-down view of the maze with a red dot highlighting the corresponding xy position. *Pixel Reach-Hard-Walls* is adapted from the Reach-v2 MetaWorld benchmark [Yu et al., 2021]. We add 4 brick walls that limit the robotic arm's ability to move freely. At the start of every episode, the robotic arm is stuck between the walls.

## 4.1 Representation Learning strategy

In this initial stage of our experiments, we set aside the intrinsic motivation component of GCRL and adopt the standard practice of sampling goals from the learned prior of the VAE, i.e. $z_g \sim \mathcal{N}(0, I)$. Our objective is to compare DRAG, which trains the VAE on data sampled from a distribution proportional to $r_\psi(x) \, p_{\pi_\theta}(x)$, with the classical RIG approach, which samples uniformly from the replay buffer, i.e. $p_{\text{rig}}(x) \propto p_{\pi_\theta}(x)$. We also include a variant taken from SKEW-FIT, where the VAE is trained on samples drawn from a skewed distribution defined as $p_{\text{skewed}}(x) \propto p_{\pi_\theta}(x)^\alpha$, with $\alpha < 0$ an hyper-parameter that acts analogously to $\lambda$ from DRAG (with $\alpha = -1/\lambda$, see Appendix C.2).

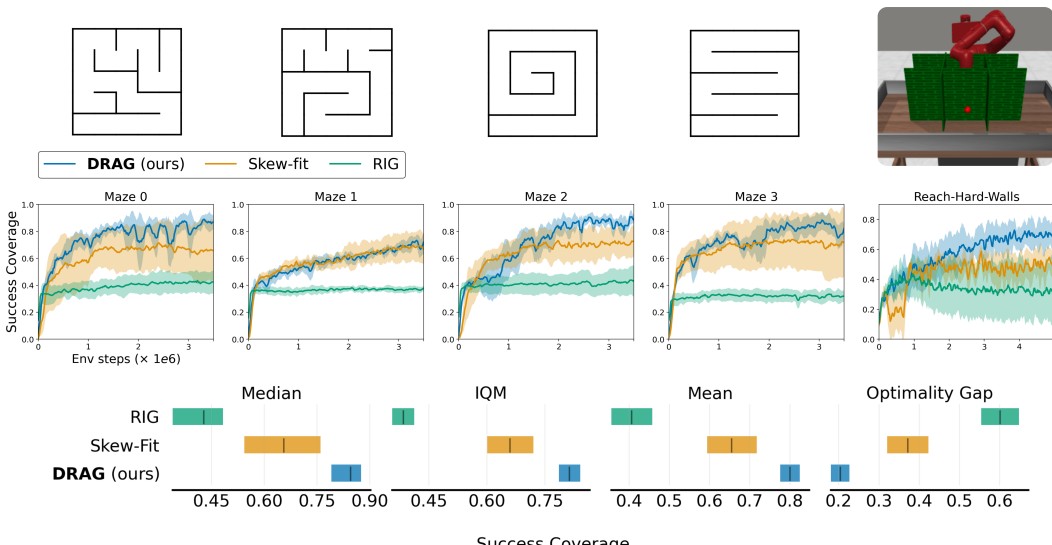

Figure 3: Evolution of the success coverage over PointMazes and Reach-Hard-Walls environments (6 seeds each) for 4M steps (shaded areas as standard deviation). Bottom: Median, Interquartile Mean, Mean and Optimality Gap of success coverage across the all runs after 4M steps. We plot these metrics and confidence intervals using the Rliable library [Agarwal et al., 2021].

**Success Coverage evaluation** Results in Figure 3 show the evolution of the success coverage over 4M steps. We see that DRAG significantly outperforms RIG and SKEW-FIT. These results corroborate that online representation learning with RIG is unable to overcome an exploration bottleneck. Therefore, a RIG agent can only explore and control a very small part of the environment. Furthermore, the success coverage of RIG is systematically capped to a certain value. SKEW-FIT is often able to overcome the exploration bottleneck but suffers from high instability, which indeed corresponds to the main drawback of non-parametric DRO, highlighted in Michel et al. [2021]. Therefore, SKEW-FIT is unable to reliably maximize the success coverage. On the other hand, DRAG is more stable due to the use of parametric likelihood ratios and is able to maximize the success coverage. Additional results and visualizations on these experiments are presented in Appendix D.1. In particular, they show a greatly better organized latent space with DRAG than with other approaches. We also show in appendix D.2 that DRAG obtains better latent representations in terms of the trustworthiness score [Venna and Kaski, 2001], which measures the preservation of

the local neighborhood structure in the input space. Metrics regarding computational runtime are reported in appendix D.6.

## 4.2 Latent Goal selection strategy

Our second question is on the impact of goal selection on the maximization of success coverage. As depicted in Figure 2b, we compare several goal selection criteria from the literature, on top of DRAG, as follows. The selection of each training latent goal is performed as follows. First, we sample a set of candidate goals $C_g = \{z_i\}_{i=1}^{N}$ from the latent prior $\mathcal{N}(0, I)$. Then, the selected goal is resampled among such pre-sampled candidates $C_g$, using one of the following strategies. Among them, **MEGA** and **LGE** only focus on exploration, **GoalGan** and **SVGG** look at the success of control:

- **MEGA** - Minimum Density selection, from [Pitis et al., 2020]: $p_{\text{mega}}(z_g) \propto \delta_c(z_g)$, where $\delta_c$ is a Dirac distribution centered on $c$, which corresponds to the code from $C_g$ with minimal density (according to a KDE estimator trained on latent codes from the buffer);

- **LGE** - Minimum density geometric sampling, from [Gallouédec and Dellandréa, 2023]: $p_{\text{lge}}(z_g) = (1-p)^{R(z_g)-1}p$, where $R(z_g)$ stands for the density rank of $z_g$ (according to a trained KDE) among candidates $C_g$, and $p$ is the parameter of a geometric distribution.

- **GoalGan** - Goals of Intermediate Difficulty selection, from [Florensa et al., 2018]: $p_{\text{goid}}(z_g) = \mathcal{U}(\text{GOIDs})$,     $\text{GOIDs} = \{z_g \in C_g | P_{min} < D(z_g) < P_{max}\}$, where $D(z_g)$ is a success prediction model, and thresholds are arbitrarily set as $P_{min} = 0.3$ and $P_{max} = 0.7$, following recommended values in [Florensa et al., 2018];

- **SVGG** - Control of goal difficulty, from [Castanet et al., 2022]: $p_{\text{skills}}(z_g) \propto \exp\left(f_{\alpha,\beta}(D(z_g))\right)$, where $D$ is a success prediction model trained simultaneously from rollouts, $f_{\alpha,\beta}$ is a beta distribution controlling the target difficulty, $\alpha = \beta = 2$, smoothly emphasize goals such that $D(z_g) \approx 1/2$.

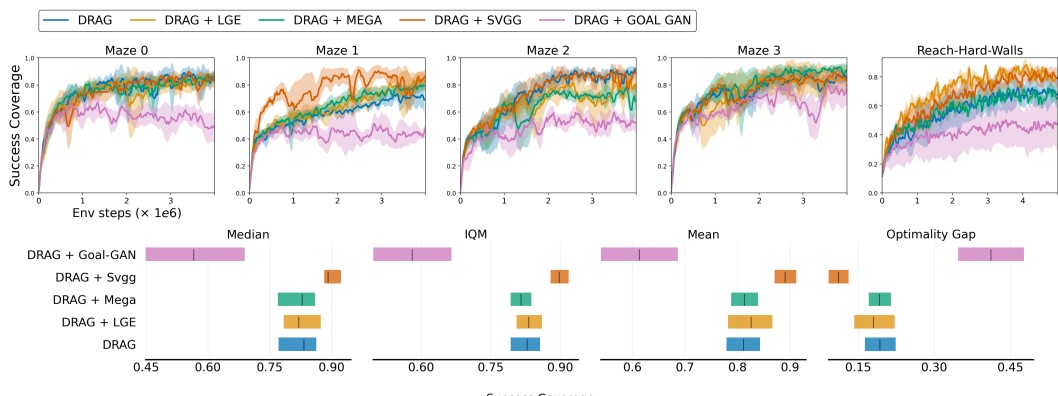

Figure 4: Impact of goal resampling on DRAG. Evolution of the success coverage for different goal sampling methods (6 seeds per run). DRAG directly uses goals sampled from the prior (i.e., same results as in figure 3), DRAG + X includes an additional goal resampling method X, taken among the four strategies: LGE, MEGA, GOALGAN or SVGG.

**Success Coverage evaluation** The success coverage results in Figure 4 reveal an interesting pattern: methods that incorporate diversity-based goal selection on top of DRAG, such as MEGA and LGE, do not lead to any improvement over the original DRAG approach using goals sampled from the prior distribution $p(z)$. This suggests a redundancy between these diversity-based criteria and our core DRO-based representation learning mechanism, which already inherently fosters exploration. Integrating a GOALGAN-like GOID selection criterion degrades DRAG's performance, likely due to its overly restrictive goal selection strategy, which hinders exploratory behaviors—hard goals sampled from the prior must be given a chance to be selected for rollouts in order to support exploration.

In contrast, the SVGG resampling distribution - which leverages the control success predictor in a smoother and more adaptive manner - significantly outperforms direct sampling from the trained prior. In general, control-based goal selection is ineffective when using classical VAE training in GCRL (as

exemplified by RIG), since goals that are not well mastered tend to be poorly represented in the latent space. However, the representation learned with DRAG enables goal selection to focus entirely on control improvement, as it ensures a more structured and meaningful latent space. Additional results on these experiments are presented in Appendix D.3.

### 4.3 Conclusion

In this work, we introduced DRAG, an algorithm leveraging Distributional Robust Optimization, to learn representation from pixel observations in the context of intrinsically motivated Goal-Conditioned agents, in online RL, without requiring any prior knowledge. We showed that by taking advantage of the DRO principle, we are able to overcome exploration bottlenecks in environments with discontinuous goal spaces, setting us apart from previous methods like RIG and SKEW-FIT.

As future work, DRAG is agnostic to the choice of representation learning algorithm, so we might consider alternatives such as other reconstruction-based techniques [Van Den Oord et al., 2017, Razavi et al., 2019, Gregor et al., 2019], or contrastive learning objectives [Oord et al., 2018, Henaff, 2020, He et al., 2020, Zbontar et al., 2021]. Besides, DRAG does not leverage pre-trained visual representations, though they could greatly improve performance on complex visual observations [Zhou et al., 2025]. In particular, we may incorporate pre-trained representations from models specific to RL tasks as VIP [Ma et al., 2022] and R3M [Nair et al., 2022] as well as general-purpose visual encoders such as CLIP [Radford et al., 2021] or DINO models [Caron et al., 2021, Oquab et al., 2024]. DRAG also opens promising avenues for discovering more principled and effective goal resampling strategies, made possible by a better anticipation of distributional shifts that previously constrained the potential of the behavioral policy.

## Acknowledgements

This work was granted access to the HPC resources of IDRIS under the allocation AD010615934 and AD011014032R2 made by GENCI. We acknowledge funding from the European Commission's Horizon Europe Framework Programme under grant agreement No 101070381 (PILLAR-robots project).

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

# A   Experimental details

## A.1   Environments

In both environments below, we transform 2D states and goals into $82 \times 82$ pixel observations.

**Pixel continuous point maze:**   This continuous 2D maze environment is taken from [Trott et al., 2019]. The action space is a continuous vector $(\delta x, \delta y) = [0, 1]^2$. Original states and goals are 2D $(x, y)$ positions in the maze and success is achieved if the L2 distance between states and goal coordinates is below $\delta = 0.15$ (only used during the evaluation of success coverage), while the overall size of the mazes is $6 \times 6$. The episode rollout horizon is $T = 50$ steps. Examples of pixel observation goals are shown in Figure 5.

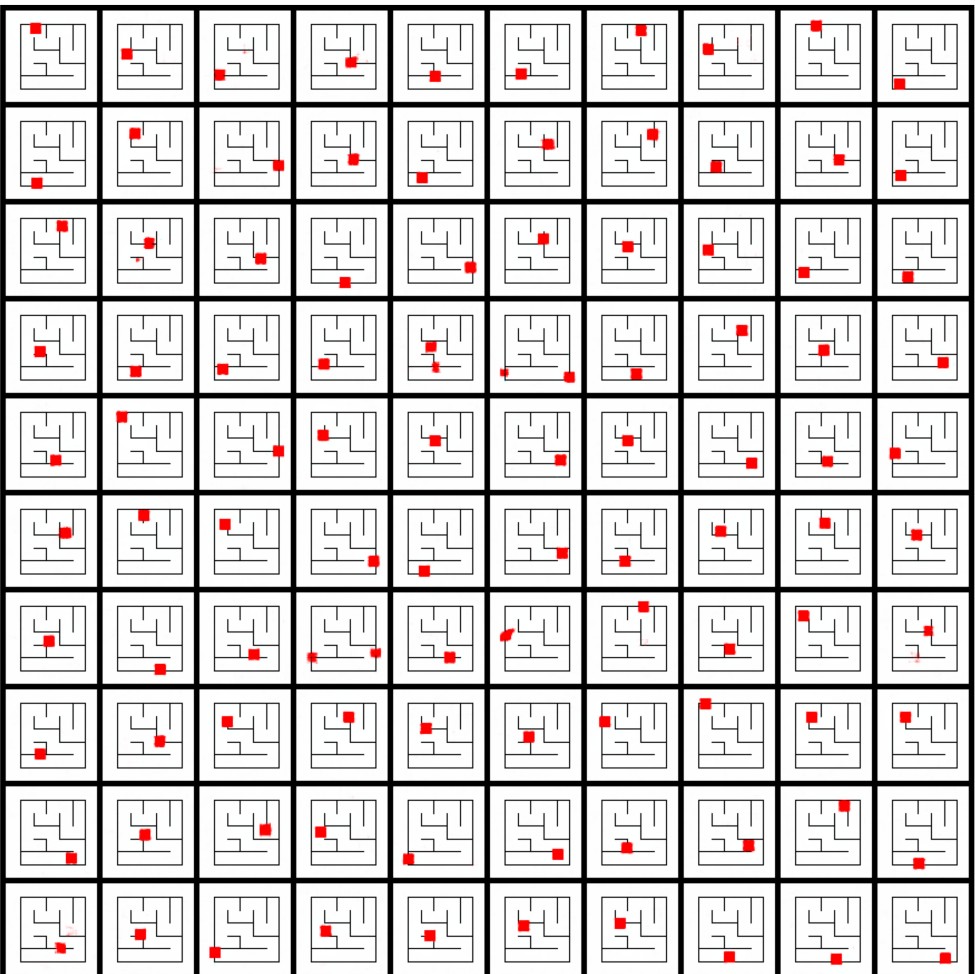

Figure 5: Example of decoded pixel goals in maze environment: we sample latent goals from the latent prior $z \sim p(z) = \mathcal{N}(0, I)$ and plot the corresponding decoded pixel goals $p_\theta(x|z)$. Images were obtained using the decoder trained with DRAG.

**Pixel Reach hard walls:**   This environment is adapted from the Reach-v2 MetaWorld benchmark [Yu et al., 2021] where the gripper is initially stuck between four walls and has to navigate carefully between them to reach the goals. The original observations are 49-dimensional vectors containing the gripper position as well as other environment variables, the actions and the goals are 3-dimensional corresponding to $(x, y, z)$ coordinates. Success is achieved if the L2 distance between states and goal

coordinates is less than $\delta = 0.1$ (only used during the evaluation of success coverage). The episode rollout horizon is $T = 300$ steps.

We transform states and goals into pixels observation using the Mujoco rendering function with the following camera configuration:

```
DEFAULT_CAMERA_CONFIG = {
    "distance": 2.,
    "azimuth": 270,
    "elevation": -30.0,
    "lookat": np.array([0, 0.5, 0]),
}
```

The walls configuration is obtained with the addition of the following bodies into the "worldbody" of the xml file of the original environment:

```
<body name="wall_1" pos="0.15 0.55 .2">
    <geom material="wall_brick" type="box" size=".005 .24 .2" rgba="0 1 0 1"/>
    <geom class="wall_col" type="box" size=".005 .24 .2" rgba="0 1 0 1"/>
</body>

<body name="wall_2" pos="-0.15 0.55 .2">
    <geom material="wall_brick" type="box" size=".005 .24 .2" rgba="0 1 0 1"/>
    <geom class="wall_col" type="box" size=".005 .24 .2" rgba="0 1 0 1"/>
</body>

<body name="wall_3" pos="0.0 0.65 .2">
    <geom material="wall_brick" type="box" size=".4 .005 .2" rgba="0 1 0 1"/>
    <geom class="wall_col" type="box" size=".4 .005 .2" rgba="0 1 0 1"/>
</body>

<body name="wall_4" pos="0.0 0.35 .2">
    <geom material="wall_brick" type="box" size=".4 .005 .2" rgba="0 1 0 1"/>
    <geom class="wall_col" type="box" size=".4 .005 .2" rgba="0 1 0 1"/>
</body>
```

Examples of pixel observation goals are shown in Figure 6.

## A.2 $\beta$-VAE

### A.2.1 Training schedule

During the first 300k steps of the agent, we train the VAE every 5k agent steps for 50 epochs of 10 optimization steps (on a dataset of 1000 inputs uniformly sampled from the buffer, divided in 10 minibatches of 100 examples). Afterward, we train it every 10k agent steps.

The following other schedules have been experimented, each getting worse average results for any algorithm:

1. During the first 300k steps of the agent, train the VAE every 10k agent steps. Afterward, train it every 20k steps.

2. During the first 100k steps of the agent, train the VAE every 5k agent steps. Afterward, train it every 10k steps.

3. During the first 100k steps of the agent, train the VAE every 2k agent steps. Afterward, train it every 5k steps.

### A.2.2 Encoder smooth update

To enhance the stability of the agent's input representations, actions are selected based on a smoothly updated version of the VAE encoder, denoted by parameters $\hat{\phi}$. This encoder is refreshed after each VAE training phase and used to produce latent states:

$$a_t = \pi_\theta(.|z_t, z_g), \quad z_t = q_{\hat{\phi}}(x_t)$$

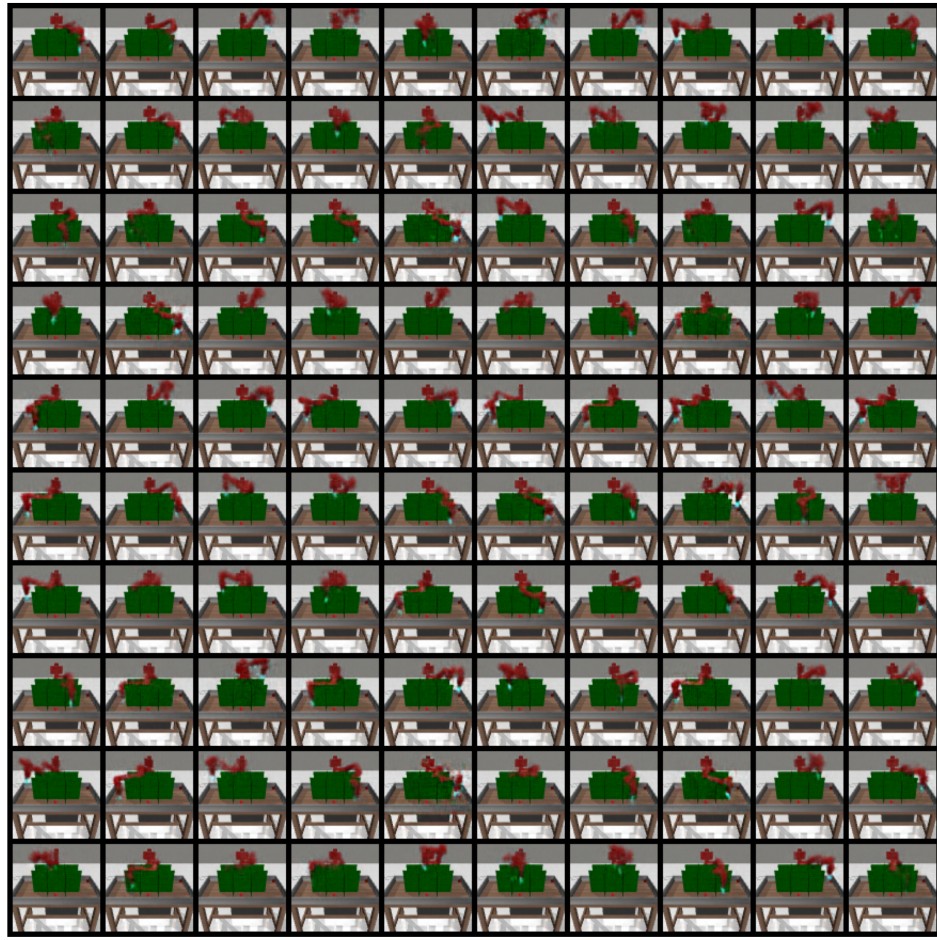

Figure 6: Example of decoded pixel goals in fetch environment: we sample latent goals from the latent prior $z \sim p(z) = \mathcal{N}(0, I)$ and plot the corresponding decoded pixel goals $p_\theta(x|z)$. Images were obtained using the decoder trained with DRAG.

Analogous to the use of a target network for $Q$-function updates in RL, the delayed encoder $q_{\hat{\phi}}$ is updated using an exponential moving average (EMA) of the primary encoder's weights $q_\phi$:

$$\hat{\phi} \leftarrow \tau \hat{\phi} + (1 - \tau)\phi \qquad (6)$$

Figure 7 illustrates how the smoothing coefficient $\tau$ influences success coverage with DRAG. We observe that $\tau = 0.05$ provides a good balance, yielding stable performance. In contrast, setting $\tau = 1$ (no smoothing) leads to less stability, while $\tau = 0.01$ results in updates that are too slow. This value was observed to provide the best average results for other approaches (i.e., RIG and SKEW-FIT). We use it in any experiment reported in other sections of this paper.

### A.3 Methods Hyper-parameters

The hyper-parameters of our DRAG algorithm are given in Table 2. Notations refer to those used in the main paper or the pseudo-code given in Algorithm 1. Hyper-parameters that are common to any approach were set to provide best average results for RIG. RIG, SKEW-FIT and DRAG share the same values for these hyper-parameters. The skewing temperature for SKEW-FIT, which is not reported in the tables below, is set to $\alpha = -1$. This value was tuned following a grid search for $\alpha \in [-100, -50, -10, -5, -1, -0.5, -0.1]$.

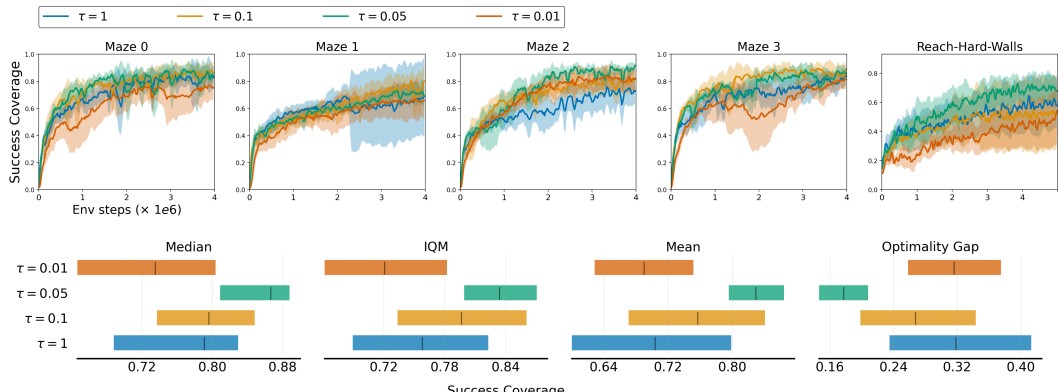

Figure 7: Impact of the exponential moving average smoothing coefficient ($\tau$ in Equation (6)) experiment on success coverage (6 seeds per run).

Table 1: Hyper-parameters used for the VAE used in the experiments (same for every approach, top) and values used specifically in DRAG, for the specification of our DRO weighter (bottom).

| VAE Hyper-Parameters | Symbol | Value |
|---|---|---|
| | | |
| $\beta$-**VAE** | | |
| Latent dim | $d$ | [Maze env :2, Fetch: 3] |
| Prior distribution | $p(z)$ | $\mathcal{N}(0, I_d)$ |
| Regularization factor | $\beta$ | 2 |
| Learning rate | $\epsilon$ | 1e-3 |
| CNN channels | | (3, 32, 64, 128, 256) |
| Dense layers | | (512,128) |
| Activation Function | | ReLu |
| CNN Kernel size | | 4 |
| Training batch size | $n$ | 100 |
| Optimization interval (in agent steps) | $freqOpt$ | 10e3 |
| Nb of training steps | $nbEpochs$ | 50 |
| Size of training buffer | $|\mathcal{R}|$ | 1e6 |
| Number of samples per epoch | $N$ | 1e3 |
| | | |
| **DRO Weighter** $r_\psi$ | | |
| Learning rate | $\epsilon$ | 2e-6 |
| Convolutional layers channels | | (3, 32, 64, 128, 256) |
| Dense layers | | (512,128) |
| Activation Function | | ReLu |
| Temperature | $\lambda$ | 0.01 |

Table 2: Off-policy RL algorithm TQC parameters

| TQC Hyper-Parameters | Value |
|---|---|
| Batch size for replay buffer | 2000 |
| Discount factor $\gamma$ | 0.98 |
| Action L2 regularization | 0.1 |
| (Gaussian) Action noise max std | 0.1 |
| Warm up steps before training | 2500 |
| Actor learning rate | 1e-3 |
| Critic learning rate | 1e-3 |
| Target network soft update rate | 0.05 |
| Actor & critic networks activation | ReLu |
| Actor & critic hidden layers sizes | $512^3$ |
| Replay buffer size ($|\mathcal{B}|$) | 1e6 |

Table 3: goal criterion hyper-parameters

| Goal criterion Hyper-parameters | Symbol | Value |
|---|---|---|
| **Kernel density Estimation for MEGA and LGE** | | |
| RBF kernel bandwidth | $\sigma$ | 0.1 |
| KDE optimization interval (in agent steps) | | 1 |
| Nb of state samples for KDE optim. | | 10.000 |
| Nb of sampled candidate goals from $p(z)$ | | 100 |
| **Agent's skill model $D_\phi$ for SVGG and GOALGAN** | | |
| Hidden layers sizes | | (64, 64) |
| Gradient steps per optimization | | 100 |
| Learning rate | | 1e-3 |
| Training batch size | | 100 |
| Training history length (episodes) | | 500 |
| Optimization interval (in agent steps) | | 5000 |
| Nb of training steps | | 100 |
| Activations | | Relu |

### A.4 Compute ressources & code assets

This work was performed with 35,000 GPU hours on NVIDIA V100 GPUs (including main experiments and ablations).

Algorithms were implemented using the GCRL library XPAG [Perrin-Gilbert, 2022], designed for intrinsically motivated RL agents.

## B  DRAG algorithm

Algorithm 1 reports the full pseudo-code of our DRAG approach. RIG and SKEW-FIT follow the same procedure, without the DRO weighter update loop (line 9 to 12), and replacing $\mathcal{L}_{\theta,\phi,r_\psi}^{\text{VAE-DRO}}(\{x_i\}_{i=1}^n)$ in line 15 by:

- RIG (classical ELBO):

$$\mathcal{L}_{\theta,\phi}^{\text{VAE}}(\{x_i\}_{i=1}^n) = \frac{n}{N} \sum_{i=1}^n \left( \frac{1}{m} \sum_{j=1}^m \log p_\theta(x_i|z_i^j) - KL(q_\phi(z|x_i)||p(z)) \right)$$

- SKEW-FIT:

$$\mathcal{L}_{\theta,\phi}^{\text{VAE-SkewFit}}(\{x_i\}_{i=1}^n) = \frac{n}{N} \sum_{i=1}^n p_{skewed}(x_i) \left( \frac{1}{m} \sum_{j=1}^m \log p_\theta(x_i|z_i^j) - KL(q_\phi(z|x_i)||p(z)) \right),$$

where $p_{skewed}(x_i)$ is the skewed distribution of SKEW-FIT, that uses an estimate $\tilde{L}_{\theta,\phi}(x_i)$ of the generative posterior of $x_i$ from the current VAE, obtained from $M$ codes sampled from $q_\phi(z|x_i)$. More details about $p_{skewed}$ are given in section C.2. For comparison, as a recall, for DRAG we take:

- DRAG:

$$\mathcal{L}_{\theta,\phi,\psi}^{\text{VAE-DRO}}(\{x_i\}_{i=1}^n) = \frac{1}{N} \sum_{i=1}^n r_\psi(x_i) \left( \frac{1}{m} \sum_{j=1}^m \log p_\theta(x_i|z_i^j) - KL(q_\phi(z|x_i)||p(z)) \right),$$

where the weighting function $r_\psi$ is defined, following equation 2, as: $r_\psi(x_i) = n \frac{\exp^{f_\psi(x_i)}}{\sum_{j=1}^n \exp^{f_\psi(x_j)}}$, for any minibatch $\{x_i\}_{i=1}^n$.

---

**Algorithm 1** *Distributionally Robust Exploration*

---

1: **Input:** a GCP $\pi_\theta$, a VAE: encoder $q_\phi(z|x)$ and smoothly updated version $q_{\hat\phi}(z|x)$, decoder $p_\phi(x|z)$, latent prior $p(z)$, DRO Neural Weighter $r_\psi$, buffers of transitions $\mathcal{B}$, reached states $\mathcal{R}$, train size $N$, batch-size $n$, number $m$ of sampled noises for each VAE training input, number $M$ of Monte Carlo samples used to estimate $\tilde{L}_{\theta,\phi}$ for each input image, temperature $\lambda$, number of optimization epochs $nEpochs$, frequence of VAE and policy optimization $freqOpt$.
2: **while** not stop **do**
3:   ▷ *Data Collection (during $freqOptim$ steps):* Perform rollouts of $\pi_\theta(.|z_t, z_g)$ in the latent space, conditioned on goals sampled from prior $z_g \sim p(z)$ or the buffer (with possible resampling depending on the goal selection strategy), and latent state $z_t = q_{\hat\phi}(x_t)$, with $x_t$ a pixel observation;
4:   Store transitions in $\mathcal{B}$, visited states in $\mathcal{R}$;
5:
6:   ▷ *Learning Representations with VAE*
7:   **for** $nEpochs$ epochs **do**
8:     Sample a train set of N states $\Gamma$ from $\mathcal{R}$
9:     **for** every mini-batch $\{x_i\}_{i=1}^n$ from $\Gamma$ **do**
10:       ▷ *DRO Weighter Update*
11:         Update weighter by one step of Adam optimizer, for the maximization problem from (4) with temperature $\lambda$, using $\tilde{L}_{\theta,\phi}$ estimated from $M$ samples from $q_\phi(z|x_i)$ for each $x_i$.
12:     **end for**
13:     **for** every mini-batch $\{x_i\}_{i=1}^n$ from $\Gamma$ **do**
14:       ▷ *Weighted VAE Update*
15:         Update encoder $q_\phi$ and decoder $p_\phi$ by one step of Adam optimizer on $-\mathcal{L}_{\theta,\phi,r_\psi}^{\text{VAE-DRO}}(\{x_i\}_{i=1}^n)$, as defined in (5), with $m$ sampling noises $(\epsilon_i^j)_{j=1}^m$ for each $x_i$.
16:         Perform smooth update of $\hat\phi$ as a function of $\phi$ according to equation (6).
17:     **end for**
18:   **end for**
19:
20:   ▷ *Agent Improvement*
21:     Improve agent with any Off-Policy RL algorithm (e.g., TQC, DDPG, SAC...) using transitions from $\mathcal{B}$;
22: **end while**

---

# C  Skew-Fit is a non-parametric DRO

In this section we show that SKEW-FIT is a special case of the non-parametric version of DRO.

## C.1 Non-parametric solution of DRO

We start from the inner maximization problem stated in (1), for a given fixed $\theta$:

$$\max_r \frac{1}{N} \sum_{i=1}^N r(x_i, y_i) l(f_\theta(x_i), y_i)) - \lambda \frac{1}{N} \sum_{i=1}^N r(x_i, y_i) \log r(x_i, y_i) \tag{7}$$

$$st \quad \frac{1}{N} \sum_{i=1}^N r(x_i, y_i) = 1.$$

From this formulation, we can see that the risk associated to a shift of test distribution can be mitigated by simply associating adversarial weights $r_i := r(x_i, y_i)$ to every sample $(x_i, y_i)$ from the training dataset, respecting $\bar{r} := \frac{1}{N} \sum_{i=1}^N r_i = 1$. This can be viewed as an infinite capacity function $r$, able to over-specify on every training data point. Equivalently to (7), we thus have:

$$\max_{(r_i)_{i=1}^N} \frac{1}{N} \sum_{i=1}^N r_i l_i - \lambda \frac{1}{N} \sum_{i=1}^N r_i \log r_i \tag{8}$$

$$st \quad \frac{1}{N} \sum_{i=1}^N r_i = 1,$$

where $l_i := l(f_\theta(x_i), y_i))$. The Lagrangian corresponding to this constrained maximization is given by:

$$\mathcal{L} = \frac{1}{N} \sum_{i=1}^N r_i l_i - \lambda \frac{1}{N} \sum_{i=1}^N r_i \log r_i - \gamma \left( \frac{1}{N} \sum_{i=1}^N r_i - 1 \right) \tag{9}$$

where $\gamma$ is an unconstrained Lagrangian coefficient.

Following the Karush-Kuhn-Tucker conditions applied to the derivative of the Lagrangian function $\mathcal{L}$ of this problem in $r_i$ for any given $i \in [[1, N]]$, we obtain:

$$\frac{\partial \mathcal{L}}{\partial r_i} = 0 \Leftrightarrow l_i - \lambda(\log r_i - 1) - \gamma = 0 \Leftrightarrow r_i = e^{\frac{l_i - \gamma}{\lambda} - 1} = z e^{\frac{l_i}{\lambda}} \tag{10}$$

with $z := e^{\frac{-\gamma}{\lambda} - 1}$.

The KKT condition on the derivative in $\gamma$ gives: $\frac{\partial \mathcal{L}}{\partial \gamma} = 0 \Leftrightarrow \frac{1}{N} \sum_{i=1}^N r_i = 1$. Combining these two results, we thus obtain:

$$\frac{1}{N} \sum_{i=1}^N r_i = \frac{1}{N} \sum_{i=1}^N z e^{\frac{l_i}{\lambda}} = 1 \Leftrightarrow z = \frac{N}{\sum_{i=1}^N e^{\frac{l_i}{\lambda}}}$$

Which again gives, reinjecting this result in Equation (10):

$$r_i = N \frac{e^{\frac{l_i}{\lambda}}}{\sum_{j=1}^N e^{\frac{l_j}{\lambda}}}$$

This leads to the form of a Boltzmann distribution, which proves the result.

## C.2 Application to GCRL with VAE and Relation to Skew-Fit

SKEW-FIT resamples training data points from a batch $\{x_i\}_{i=1}^n$ using a skewed distribution defined, for any sample $x$ in that batch, as:

$$p_{\text{skewed}}(\mathbf{x}) \triangleq \frac{1}{Z_\alpha} w_{t,\alpha}(\mathbf{x}), \tag{11}$$

$$Z_\alpha = \sum_{i=1}^n w_{t,\alpha}(\mathbf{x}_i),$$

where $w_{t,\alpha}$ is an importance sampling coefficient given as:

$$w_{t,\alpha}(x) := p_{\theta,\phi}(x)^\alpha, \quad \alpha < 0, \tag{12}$$

with $p_{\theta,\phi}(x)$ the generative distribution of samples $x$ given current parameters $(\theta, \phi)$.

Applied to a generative model defined as a VAE, we have:

$$p_{\theta,\phi}(x) = \mathbb{E}_{z \sim q_\phi(z|x)} \frac{p(z)p_\theta(x|z)}{q_\phi(z|x)} dz,$$

where $p(z)$ is the prior over latent encodings of the data $x$, $p_\theta(x|z)$ is the likelihood of $x$ knowing $z$ and $q_\phi(z|x)$ the encoding distribution of data points. As stated in Section 3.1, this can be estimated on a set of $m$ samples for each data point using the log-approximator: $\tilde{\mathcal{L}}_{\theta,\phi}(x) = \log \sum_{j=1}^M \exp(\log p_\theta(x|z^j) + \log p_\theta(z^j) - \log q_\phi(z^j|x)) - \log(M)$.

Thus, this is equivalent as associating any $i$ from the data batch with a weight $r_i$ defined as:

$$\begin{aligned} r_i := p_{\text{skewed}}(\mathbf{x_i}) &= \frac{1}{Z_\alpha} e^{\alpha \tilde{\mathcal{L}}_{\theta,\phi}(x_i)}, \tag{13} \\ Z_\alpha &= \sum_{j=1}^n e^{\alpha \tilde{\mathcal{L}}_{\theta,\phi}(x_j)}, \end{aligned}$$

for any $\alpha < 0$. Setting $\alpha = -\frac{1}{\lambda}$, we get $p_{\text{skewed}}(\mathbf{x_i}) \propto e^{-\tilde{\mathcal{L}}_{\theta,\phi}(x_i)/\lambda}$, for any temperature $\lambda > 0$. Reusing the result from Section C.1, this is fully equivalent to the analytical closed-form solution of DRO when applied to $-\tilde{\mathcal{L}}_{\theta,\phi}(x_i)$ as we use in our DRO-VAE approach. Using $p_{skewed}$ with $\alpha = -\frac{1}{\lambda}$ for weighting training points of a VAE thus exactly corresponds to the non-parametric version our DRAG algorithm.

# D    Aditional results

## D.1    Visualization of Learned Latent Representations

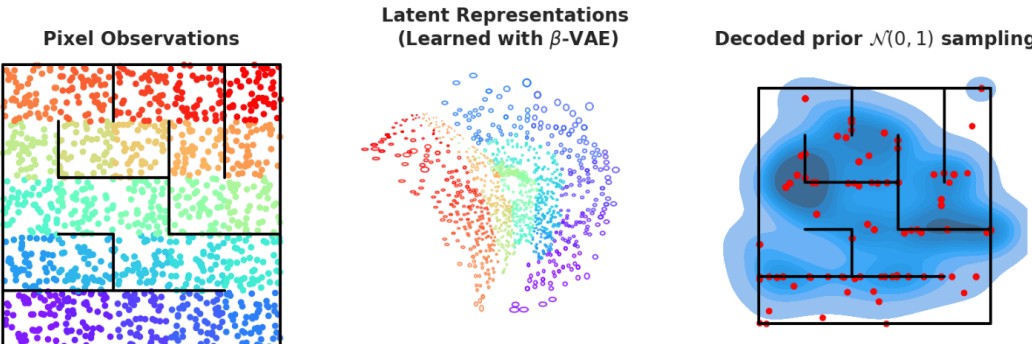

Figure 8: Learned Representation of DRAG after 1 million training steps in Maze 0. **Left**: every colored dot corresponds to the pixel observation $x$ of its specific xy coordinates. **Middle**: Every pixel observation $x$ on the left is processed by the VAE encoder to get the learned latent posterior distribution $q_\phi(z|x) = \mathcal{N}(z|\mu_\phi(x), \sigma_\phi(x))$. Colored ellipsoids correspond to these 2-dimensional Gaussian distributions. **Right**: we sample latent goals from the latent prior $z \sim p(z) = \mathcal{N}(0, I)$ and we decode the corresponding pixel observations $p_\theta(x|z)$ (red dots correspond to the xy coordinates of the pixel observations).

Figure 8 presents our methodology to study latent representation learning. We uniformly sample data points in the maze and process them iteratively from 2D points to pixels, then from pixels to the latent code of the VAE. Using the same color for the source data points and the latent code, this process allows us to visualize the 2D latent representation of the VAE in the environment (Figure 8 left and

middle). In addition, to get a sense of what part of the environment is encoded in the latent prior, we sample latent codes from $p(z)$ and plot the 2D coordinates of the decoded observations using $p_\theta(x|z)$, which corresponds to the red dots. The blue distribution corresponds to a KDE estimate fitted to the red dots.

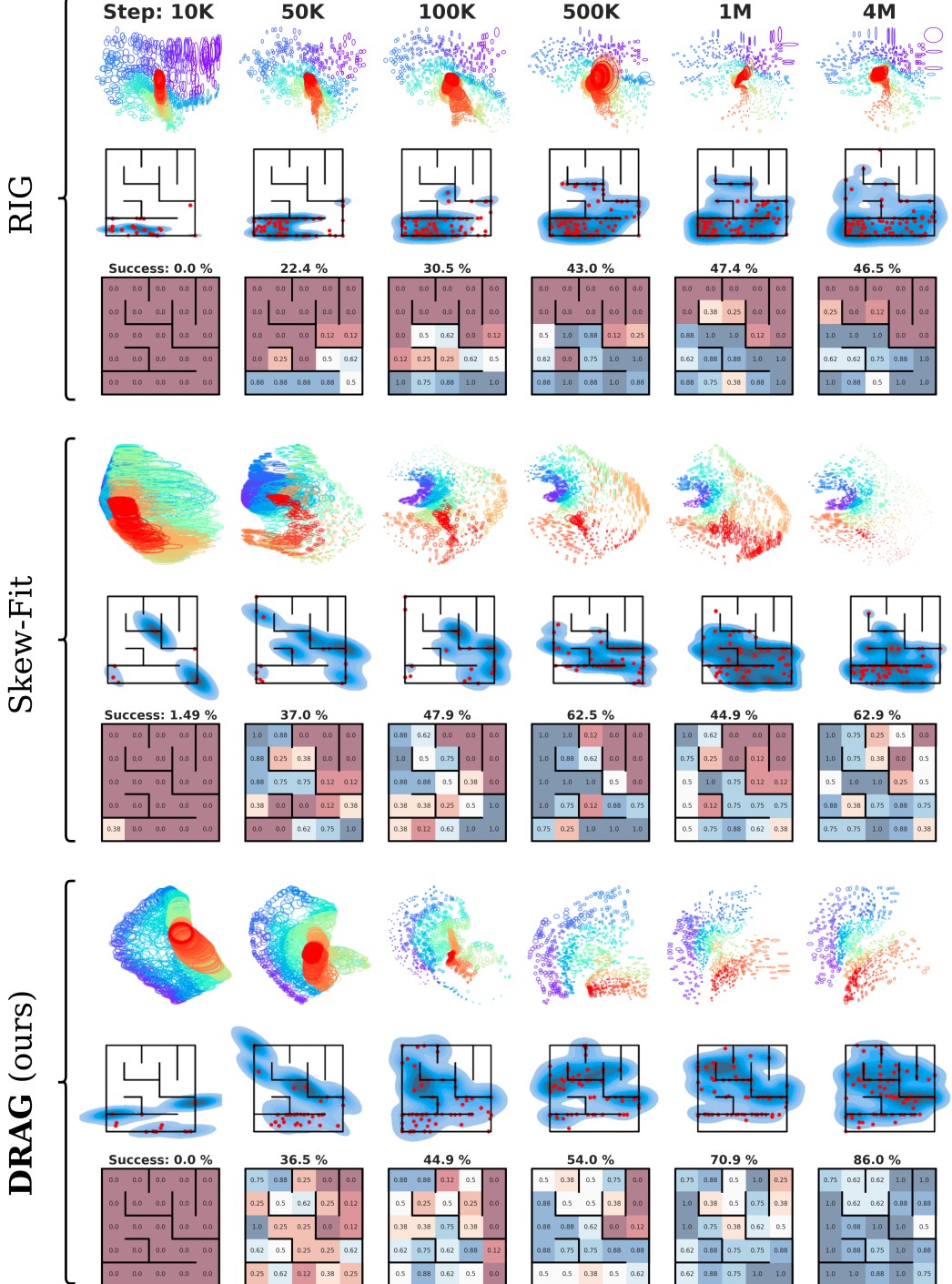

Figure 9: **First row** of each method: evolution of learned representations. **Second row** of each method: evolution of the intrinsic goal distribution when sampling from the latent prior $p(z) = N(0, 1)$. **Third row** of each method: evolution of the success coverage. (See Figure 8 for details on how we obtain these plots).

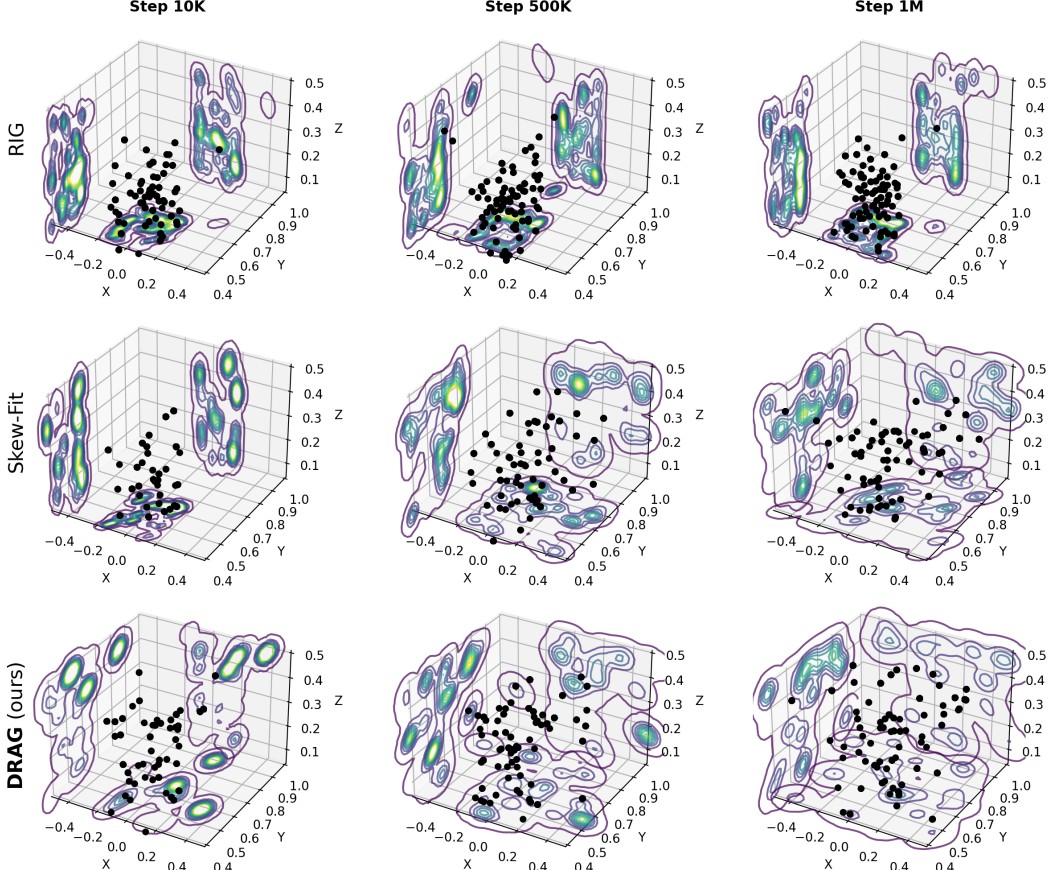

Figure 10: Evolution of the prior distribution in the Fetch environment for the DRAG, SKEW-FIT, and RIG methods: We sample latent goals from the latent prior $z \sim p(z) = \mathcal{N}(0, I)$ and we decode the corresponding pixel observations $p_\theta(x|z)$ (black dots correspond to the 3D xyz coordinates of the pixel observations).

In order to gain a deeper insight into the performance of RIG, SKEW-FIT, and DRAG, we show in Figure 9 the parallel evolution of the prior sampling $z \sim \mathcal{N}(0, I)$, and the corresponding learned 2D representations for the maze environment.

One can clearly see that RIG is stuck in an exploration bottleneck (which in this case corresponds to the first U-turn of the maze): the VAE cannot learn meaningful representations of poorly explored areas (red part of the maze in Figure 9). As a consequence, the prior distribution $p(z)$ only encodes a small subspace of the environment. On the other hand, SKEW-FIT and DRAG manage to escape these bottlenecks and incorporate an organized representation of nearly every area of the environment, with the difference that DRAG is more stable and therefore reliably learns well organized representations.

In order to quantify the evolution of latent representations to highlight the differences in terms of latent distribution dynamics between RIG, SKEW-FIT, and DRAG, we introduce the following measurement:

$$\forall t = 1...T, \quad d_t(\mathbf{x}) \triangleq \frac{1}{n} \sum_{i=1}^{n} \| \mu_{\phi^t}(x_i) - \mu_{\phi^{t-1}}(x_i) \|, \tag{14}$$

where $\mathbf{x} = \{x_i\}_{i=1}^{n}$ is a batch of pixel observations uniformly sampled from the environment state space using prior knowledge (only for evaluation purposes). With this metric, we measure the evolution of the embedding of every point $x_i$, using the movement of the expectation $\mu_\phi(x_i)$ from the

latent posterior distribution $q_\phi(z|x_i) = \mathcal{N}(z|\mu_\phi(x_i), \sigma_\phi(x_i))$, throughout updates of VAE parameters $\phi$.

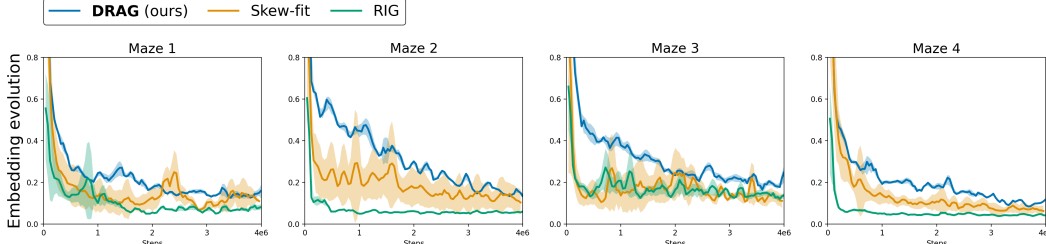

Figure 11: Evolution of the embedding over 4 different PointMazes (6 seeds each) for 4M steps (shaded areas correspond to standard deviation). Every point corresponds to the shift of representation between step $t$ and step $t+1$ of VAE training: $\frac{1}{n}\sum_{i=1}^{n} \parallel \mu_{\phi^{t+1}}(x_i) - \mu_{\phi^t}(x_i) \parallel$. For every pixel observation $x_i$ and timestep $t$, we have $q_{\phi^t}(z|x_i) = \mathcal{N}(z|\mu_{\phi^t}(x_i), \sigma_{\phi^t}(x_i))$. We compute representation shifts between $t$ and $t+1$ every $40,000$ training steps.

Figure 11 shows that the embedding movement $d$ of DRAG is higher and less variable across seeds, indicating that the learned representations evolve more consistently. Meanwhile, the VAE training process of SKEW-FIT is prone to variability, and the evolution of the embedding in RIG is close to null after a certain number of training steps.

## D.2 Representation quality

We assess the quality of the obtained representations in term of the trustworthiness score [Venna and Kaski, 2001], which measures to what extent the local neighborhood structure in the input space is preserved in the latent space. Higher trustworthiness indicates better preservation of task-relevant information across the encoding. We computed this trustworthiness metric (with 5 nearest neighbors) using a batch of 1K uniformly sampled observations from the valid state space, identical to the batch used for success coverage. Our experiments show that DRAG consistently improves this score over baseline encoders across environments.

Table 4: Mean trustworthiness of obtained representations across environments after 4M training steps.

| Methods | Trustworthiness [Venna & Kaski, 2001] |
|---------|---------------------------------------|
| **DRAG** | **98.7%** |
| **SKEWFIT** | 93.3% |
| **RIG** | 88.5% |

## D.3 Ablations

We study the impact of the main hyper-parameters, namely $\lambda$ and $M$.

### D.3.1 Impact of $\lambda$

Figure 12 illustrates the effect of varying the regularization parameter $\lambda$ on the performance of DRAG. As discussed in the main paper, lower values of $\lambda$ bias the training distribution to emphasize samples from less covered regions of the state space. Conversely, higher values of $\lambda$ lead to flatter weighting distributions across batches, eventually resembling the behavior of a standard VAE (as used in RIG) when $\lambda$ becomes very large. In fact, setting $\lambda = \infty$ makes DRAG behave identically to RIG.

The reported results show that DRAG achieves the highest success coverage for $\lambda$ values between 10 and 100, with a slight edge at $\lambda = 100$. This range represents a good trade-off: too small a $\lambda$ can lead to unstable training, where the model places excessive weight on underrepresented samples; too large a $\lambda$ leads to overly strong regularization toward the marginal distribution $p(x)$, limiting generalization, and hence exploration.

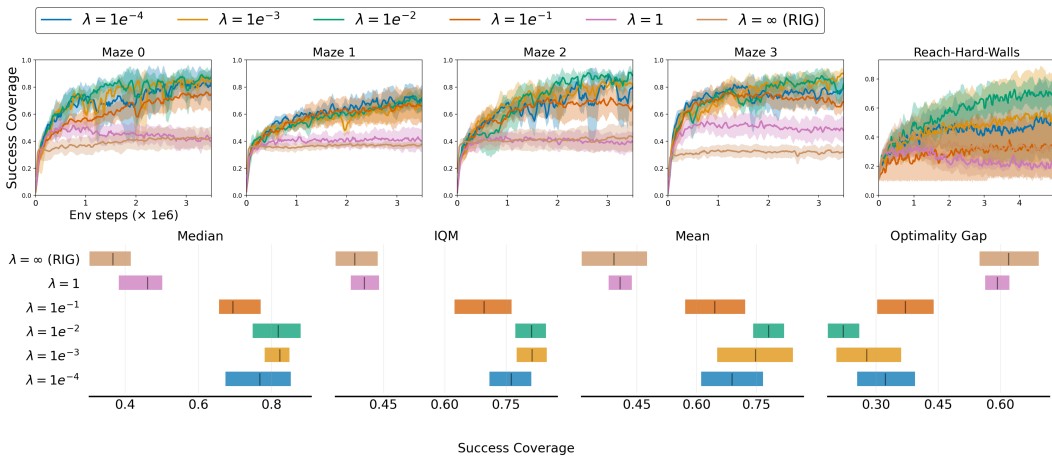

Figure 12: Impact of the regularization parameter $\lambda$ on the performances of DRAG (6 seeds per run). Results obtained with goals directly selected from the prior (as in Section 4.1). Note that $\lambda = \infty$ comes down to the RIG approach, as weights converge to a constant (over-regularization).

For comparison, SKEW-FIT performs best at $\alpha = -1$, which corresponds to $\lambda = 1$ in the DRO (non-parametric) formulation (see Section C for theoretical equivalences). This much lower value reflects a key difference: SKEW-FIT relies on pointwise estimations of the generative posterior, while DRAG uses smoothed estimates provided by a neural weighting function. As a result, SKEW-FIT requires less aggressive skewing to avoid instability.

### D.3.2 Impact of $M$ (number of samples for the estimation of $\tilde{L}_{\theta,\phi}$ )

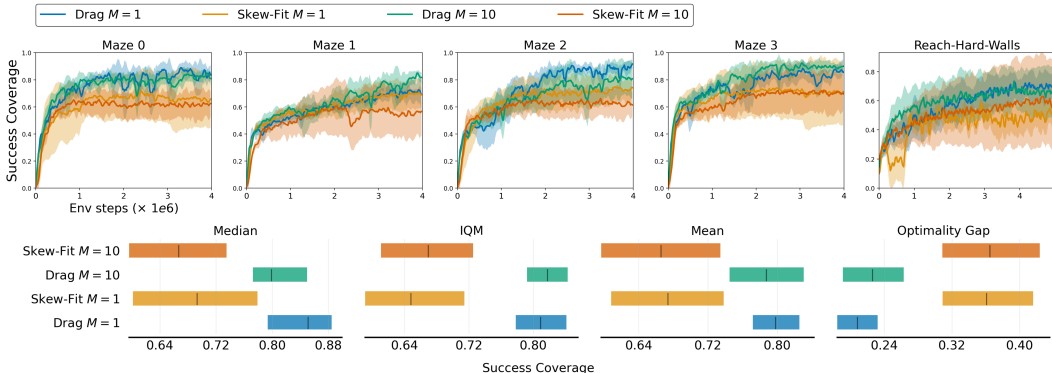

Figure 13: Impact of the number of samples $M$, used for the estimation of $\tilde{L}_{\theta,\phi}$ in DRAG and SKEW-FIT (6 seeds per run). Results obtained with goals directly selected from the prior (as in section 4.1).

Both SKEW-FIT and DRAG need to estimate the generative posterior of inputs in order to build their VAE weighting schemes. This estimator, denoted in the paper as $\tilde{L}_{\theta,\phi}(x)$ for any input $x$, is obtained via Monte Carlo samples of codes from $q_\phi(z|x)$. The number $M$ of samples used impacts the variance of this estimator. The higher $M$ is, the more accurate the estimator is, at the cost of an increase of computational resources ($M$ samples means $M$ likelihood computations through the decoder). This section inspects the impact of $M$ on the overall performance. Figure 13 presents the results for SKEW-FIT and DRAG using $M = 1$ (as in the rest of the paper) and $M = 10$. While one might expect more accurate estimates of $\tilde{L}_{\theta,\phi}(x)$ with $M = 10$, this improvement does not translate into better success coverage for the agent. According to the results, the value of $M$ does not appear to significantly impact the agent's performance for either algorithm. In fact, on average, increasing $M$ even slightly decreases success coverage.

This is a noteworthy finding, as it suggests that the improved stability of DRAG compared to SKEW-FIT is not due to more accurate pointwise likelihood estimation (which could benefit DRAG through the inertia introduced by using a parametric predictor), but rather due to greater spatial smoothness. This smoothness arises from the $L$-Lipschitz continuity of the neural network: inputs located in the same region of the visual space are assigned similar weights by DRAG's neural weighting function. In contrast, SKEW-FIT may overemphasize specific inputs, with abrupt weighting shifts, particularly when those inputs are poorly represented in the latent space, despite being located in familiar visual regions.

## D.4 RIG+Goal selection criterion

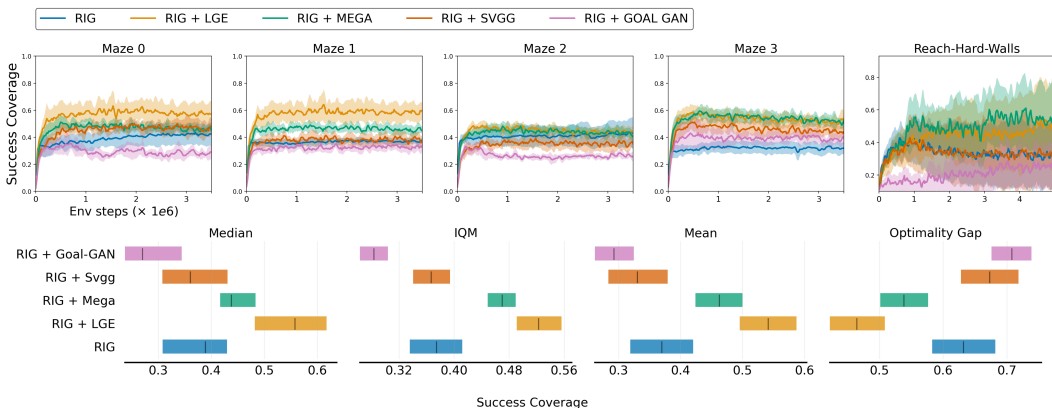

Figure 14: Impact of goal resampling with classical (unbiased) VAE training, as in RIG. Evolution of the success coverage for different goal sampling methods (6 seeds per run). RIG directly uses goals sampled from the prior (i.e., same results as RIG in figure 3), RIG + X includes an additional goal resampling method X, taken among the four strategies: LGE, MEGA, GOALGAN or SVGG. This figure presents the same experiment as in Figure 4, but using a standard VAE instead of our proposed DRO-VAE.

Figure 14 presents the results of combining the RIG representation learning strategy (i.e. without biasing VAE training) with a goal selection criterion, following the same experimental setup as in Figure 2b. These results highlight two key insights.

First, the results show that the exploration limitations inherent to the RIG strategy cannot be effectively addressed by improved goal selection alone. Even the best-performing combination (RIG + LGE) achieves less than 60% success coverage on average across environments, while DRAG alone reaches 80%. This highlights the critical role of DRAG's representation learning capability in overcoming exploration bottlenecks in complex environments. When relying solely on the latent space of a standard VAE, sampling —even when guided by intrinsic motivation— remains limited to regions already well-represented in the training data. The model cannot generate goals in poorly explored areas, as no latent codes exist that decode to such states.

Second, we observe a reversal in the relative effectiveness of goal selection criteria compared to the DRAG experiments in Figure 2b. In the case of RIG, both LGE and MEGA outperform SVGG, which contrasts with the pattern observed with DRAG. This can be explained by the fact that RIG is inherently limited in its ability to explore, and thus benefits more from goal selection strategies that explicitly promote exploration. In contrast, strategies based on intermediate difficulty, such as SVGG, are less effective when the agent is confined to a limited region of the environment. Latent codes associated with intermediate difficulty typically decode to well-known states, while those corresponding to poorly explored areas often lead to posterior distributions with higher variance. As a result, the latter are more likely to be classified as too difficult and filtered out. Therefore, these strategies tend to foster learning around familiar areas, without actively pushing the agent toward under-explored or novel regions that are critical for improving coverage. This type of goal selection can therefore only be effective when built on top of representations—such as the one learned by DRAG — that are explicitly encouraged to include marginal or rarely visited states.

### D.5 Image size study

To complement our experiments, we ran the three methods (i.e. RIG, SKEW-FIT and DRAG) on all environments with a higher image resolution (128x128 instead of 82x82 in the main experiments). We observe that all methods degrade slightly with higher resolution, but DRAG retains a clear advantage.

Table 5: Mean success coverage across maze and robotic environment for 128x128 pixels observations (success coverage for 82x82 is given for comparison).

| Methods | 1M | 2M | 3M | 4M steps |
|---|---|---|---|---|
| **DRAG** | **42%** | **58%** | **71%** | **79%** |
| 82x82 comparison | 46% | 66% | 74% | 81% |
| **SKEWFIT** | 28% | 44% | 57% | 65% |
| 82x82 comparison | 33% | 51% | 59% | 66% |
| **RIG** | 22% | 29% | 35% | 38% |
| 82x82 comparison | 28% | 34% | 39% | 42% |

### D.6 Runtime

To assess the impact of the overhead induced by the additional likelihood estimation (for SKEWFIT and DRAG compared to RIG) and the use of a neural weighter (for DRAG), Table 6 reports mean execution runtime for Maze and Metaworld environments averaged over 6 seeds on each environment. We observe that DRAG and SKEWFIT require on average about 200 seconds for 20k steps, against 176 seconds for RIG. This slight overhead is negligible compared to the time spent in environment interactions. This is because we only use one sample (i.e. ) for the likelihood estimation in DRAG and SKEWFIT (see appendix D.3.2 for a comparative study with more samples, which does not impact the results of both methods). Also, no notable difference in runtime is observed when comparing DRAG to SKEWFIT.

Table 6: Mean execution runtime on V100 GPU (32GB) for MAZE and METAWORLD environments.

| Method | Time per 20K steps (s) | Time for 4M steps (s) |
|---|---|---|
| **DRAG** | 211 | 42,200 (11.72 h) |
| **SKEWFIT** | 208 | 41,600 (11.55 h) |
| **RIG** | 176 | 35,200 (9.77 h) |

To better link performance to runtime, Table 7 also shows success coverage over wall-clock time (82×82, same runs as Fig. 3):

Table 7: Success coverage metric per hour.

| Hour | DRAG (%) | SKEWFIT (%) | RIG (%) |
|---|---|---|---|
| 1 | 34 | 31 | 26 |
| 3 | 45 | 42 | 28 |
| 5 | 60 | 50 | 32 |
| 7 | 70 | 58 | 36 |
| 9 | 77 | 63 | 40 |
| 11 | 81 | 66 | 42 |

## E Limitations

The main limitations of our work are the following.

**Latent space reward definition**    While our study makes progress on learning representations online and generating intrinsic goals from high-dimensional observations, it does not address how to measure when a goal has truly been achieved. Throughout our experiments, we relied on a simple sparse reward $r_t = \mathbb{1}\big[||z_{x_t} - z_g||_2 < \delta\big]$ which, although common in goal-conditioned RL, sidesteps the challenges of defining a dense feedback signal. In particular, the Euclidean metric used in dense rewards often fails to reflect the true topology of the environment and can mislead the agent.

**Representation Learning algorithm**    Our DRO-based approach is agnostic to the choice of representation learning algorithm, suggesting future work should benchmark alternatives such as other reconstruction-based techniques [Van Den Oord et al., 2017, Razavi et al., 2019, Gregor et al., 2019], or contrastive learning objectives [Oord et al., 2018, Henaff, 2020, He et al., 2020, Zbontar et al., 2021].

Notably, contrastive methods [Stooke et al., 2021] and Forward-Backward approaches [Touati and Ollivier, 2021] aim to incorporate dynamics by bringing temporally adjacent states closer or by modeling universal rewards. However, these methods generally assume access to transitions from a representative part of the environment. To our knowledge, they do not include any explicit mechanism to avoid collapse onto narrow parts of the state space—a critical issue in hard exploration settings without expert priors.

Addressing this limitation is precisely the aim of our DRO-based reweighting, which promotes state space coverage even in sparse reward regimes. While our implementation focuses on $\beta$-VAEs for interpretability and disentanglement, our DRO framework is general and not tied to a specific representation architecture.

In fact, any representation learning method trained from replay buffer samples using a loss $\ell(f_\theta(x))$ can be reweighted using the DRO objective:

$$\min_\theta \max_r \frac{1}{N} \sum_{i=1}^{N} r(x_i)(\ell(f_\theta(x_i))) - \lambda r(x_i) \log r(x_i) \qquad \text{s.t.} \quad \frac{1}{N} \sum_{i=1}^{N} r(x_i) = 1$$

This includes diffusion-based decoders, contrastive losses (e.g., InfoNCE), and temporal-difference-based objectives (e.g., in Forward-Backward RL). For generative losses, as in VAEs or diffusion models, the optimization must rely on likelihood estimates (i.e., $\log p_{\theta,\phi}(x)$ ), similarly to our alternate optimization strategy described in (4).

**Leveraging Pre-trained Representations**    Our study did not leverage pre-trained visual representations, that could greatly improve performance on complex visual observations as demonstrated in [Zhou et al., 2025]. In particular, future work should explore incorporating into our setting pre-trained representations from models specific to RL tasks as VIP [Ma et al., 2022] and R3M [Nair et al., 2022] as well as general-purpose visual encoders such as CLIP [Radford et al., 2021] or DINO models [Caron et al., 2021, Oquab et al., 2024].

