# OpenReview forum: "Imagine Beyond ! Distributionally Robust Autoencoding for State Space Coverage in Online Reinforcement Learning"
_NeurIPS.cc/2025/Conference — NeurIPS 2025 poster_

### Official Review · Reviewer_pMkP · 2025-06-30

**Clarity:** 3
**Significance:** 2
**Originality:** 3
**Rating:** 4
**Confidence:** 4

**Summary:**

This paper proposes a method to improve representation learning in online goal-conditioned reinforcement learning, particularly in complex visual environments. The key idea is to guide the learning process toward better state space coverage by integrating principles from distributionally robust optimization into a VAE framework. By adaptively reweighting training samples, the method encourages learning from underrepresented states, leading to more effective exploration and goal-reaching capabilities. The approach is compatible with existing GCRL frameworks and demonstrates improved performance on tasks requiring broad exploration, without relying on prior knowledge or pretraining.

**Questions:**

1. Can the authors clarify how reweighting VAE training samples directly addresses the distributional shift problem? While the method uses DRO-inspired sample weighting, the theoretical or empirical justification for why this resolves or leverages distributional shift is not sufficiently discussed.
2. Extending experiments and discussions about the approach beyond the β-VAE setup could help demonstrate its general applicability and broaden its impact.
3. Could the authors provide additional evaluation metrics to assess the quality of the learned representations? For example, visualizations of the latent space or metrics such as clustering quality, mutual information could help support the claim of improved representation learning.
4. It would be helpful to include ablation hyper-parameters analysis to better understand the contribution of each component in DRAG.
5. The paper would benefit from a clearer explanation of how the proposed method directly addresses the challenges outlined in the introduction. Currently, the transition from the problem motivation to the technical formulation of DRAG lacks sufficient intermediate discussion. Rather than immediately diving into optimization objectives and formulas, it would be helpful to first provide illustrative examples that connect the theoretical design to the practical issues raised earlier. This could significantly improve readability and help the reader understand why the proposed weighting mechanism is a suitable and effective solution to the stated problem.

**Ethical Concerns:**

["NO or VERY MINOR ethics concerns only"]

**Final Justification:**

The rating has been updated based on the authors' rebuttal.

**Limitations:**

Yes

**Quality:**

2

**Strengths And Weaknesses:**

Strength:
1. The paper identifies a critical limitation in prior GCRL methods, that VAE-based representations tend to overfit to frequently visited states. By addressing this issue, the proposed approach enables broader and more effective exploration of the environment.
2. The authors introduce a principled method that integrates distributionally robust optimization into VAE training. This design effectively mitigates distributional bias and encourages learning from underexplored states, leading to improved coverage of the state space.
3. The method is evaluated across two environments, demonstrating its generality and robustness across different task types.

Weakness:
1. The paper emphasizes the importance of addressing distributional shift in the introduction. However, the proposed method focuses on reweighting training samples in the VAE using DRO. The connection between this sample reweighting and the mitigation or utilization of distributional shift remains underexplained. It is not fully clear why optimizing VAE weights in this manner would directly resolve the shift-related issues discussed earlier.
2. While the method is evaluated on two distinct environments, the paper does not sufficiently explore the generalizability of the approach to broader representation learning paradigms. In particular, it remains unclear whether the proposed distributionally robust framework could be extended to alternative representation learning methods such as contrastive learning or diffusion-based models, which are increasingly used in modern reinforcement learning.
3. The evaluation relies solely on success coverage, without additional analysis of representation quality (e.g., latent space visualization, clustering metrics). Moreover, the experimental design lacks ablation studies or hyper-parameters analyses on key components. This limits the understanding of the contribution and robustness of individual components in the proposed framework.

---

> ### Author Rebuttal · Authors · 2025-07-30
>
> Many thanks for your thorough and constructive feedback. We appreciate the opportunity to clarify the key motivations, theoretical underpinnings, and empirical design of our work. Below, we address each of your concerns in detail.
>
> ***Weakness 1, Question 1, Question 5: Clarification of the connection between distributional shift and DRO-based sample reweighting***
>
> Thank you for the insightful comments. While Section 2.4 provides an  explanation, we agree that the connection between DRO and distributional shift could be more clearly emphasized and that we should do so earlier. We will add a clarifying sentence in the introduction.
>
> In short, Distributionally Robust Optimization (DRO) is a family of approaches specifically designed to deal with distributional shifts that can arise at test time. This was originally applied to supervised ML tasks (e.g., classification), by optimizing performance under worst-case distributions within a certain neighborhood of the empirical data distribution, by reweighting training samples. This naturally lends to anticipate test-time distributions that differ from those encountered during training, by giving more emphasis on training data points located on less populated areas of the feature space (we refer the reviewer for instance to [Rahimian and Mehrotra, 2019] or [Michel et al., 2022] for further justification on this connection). The strength of this robustness is controlled by a KL-regularized constraint (via λ), which interpolates between focusing on worst-case regions (low λ) and recovering the standard objective (high λ).
>
> In the VAE setting, applying DRO leads to reweighting samples with high reconstruction loss—typically those in sparse or high-variance latent regions—thus preventing the model from collapsing to dominant modes. While DRO has never been used in generative modeling (since VAEs are usually employed to reproduce a given data distribution, not a larger distribution around it), we argue it is especially valuable in online representation learning for GCRL, where the data distribution shifts continuously and lacks clear support boundaries.
>
> This shift creates a “vicious circle”: poor representation for underrepresented states induce narrow policies, which in turn induce narrow data. Our method breaks this loop by reweighting toward underrepresented states using the VAE’s own generative distribution—effectively steering the encoder toward better support for future exploration.
>
> We agree that a visual illustration (that we already have, but  unfortunately we cannot include it in this rebuttal due to new NeurIPS policies) would clarify this intuition, and will include it in the final version.
>
> ***Weakness 2 & Question 2: On generalization to other representation learning paradigms***
>
> We agree this is an important direction. While the paper focuses on β-VAEs due to their interpretability and controlled disentanglement properties, the DRO-based reweighting principle is not tied to any specific representation learning architecture.
>
> In fact, following the DRO principle, any method that learns representations from replay buffer samples using a loss function can be recast as:
>
> $\min_{\theta} \max_{r} \frac{1}{N} \sum_{i=1}^N r(x_i) (\ell(f_\theta(x_i))) - \lambda  r(x_i) \log r(x_i) \qquad \text{s.t.   }
> \frac{1}{N} \sum_{i=1}^N r(x_i) = 1,$
>
> where the sum is computed on $N$ samples $x_i$ from the buffer, and $\ell(f_\theta(x_i))$ corresponds to a quantity to be minimized for outputing $f_\theta(x_i)$ for sample $x_i$. For Forward-Backward approaches [Touati and Ollivier, 2021], this loss stands as a Bellman temporal difference on the successor measure. For state diffusion methods, this corresponds to a negative log-likelihood of the diffusion decoder output. Note that, similarly to our DRO-VAE, as diffusion models imply a lower bound, $r$ must also be optimized from an estimation of the true likelihood of $\tilde{ L_{\theta,\phi}}(x)\approx \log{p_{\theta,\phi}(x)}$ rather than the ELBO, in an alternate optimization scheme as we propose in Eq. (4). Other extensions, using contrastive learning to include dynamics in the representation, could incorporate an InfoNCE term alongside reconstruction in a VAE-like setting, to bring temporally adjacent states closer in latent space (as in [Stooke et al., 2021]) while encouraging the representation to cover the full state distribution from the buffer. We leave this for future work.
>
>
> [Touati and Ollivier, 2021] Touati, Ahmed, and Ollivier, Yann. "Learning one representation to optimize all rewards." Advances in Neural Information Processing Systems 34 (2021): 13-23.
>
> [Stooke et al., 2021] Stooke, Adam, et al. "Decoupling representation learning from reinforcement learning." International conference on machine learning. PMLR, 2021.
>
> ***Weakness 3, Question 3: On representation quality metrics and additional evaluation***
>
> We appreciate the reviewer’s suggestion. While we did not report clustering metrics or mutual information, we provide several visualizations and analyses of the latent space evolution over training in the appendix (please refer to the submitted supplementary material). Specifically:
>
> - **Figure 9** shows the evolution of the learned representation, alongside the intrinsic goal distribution and the success coverage, for DRAG, SKEWFIT, and RIG in 2D Maze tasks;
> - **Figure 10** depicts the evolution of the learned prior distribution in the Fetch environment for the same three methods;
> - **Figure 11** presents the magnitude of representation change over training, measured through latent distance evolution.
>
> From these figures, we observe that DRAG consistently yields more structured and well-distributed representations, leading to better state space coverage. Additionally, DRAG's representation evolves more steadily and with less variability across seeds. In contrast, SKEWFIT displays unstable training dynamics, while RIG exhibits almost no change in the latent space beyond a certain training point.
>
> To complement these results from the paper, we followed the reviewer's suggestion by assessing the obtained representations in term of the trustworthiness score [Venna & Kaski, 2001], which measures to what extent the local neighborhood structure in the input space is preserved in the latent space. Higher trustworthiness indicates better preservation of task-relevant information across the encoding. We computed this trustworthiness metric (with 5 nearest neighbors) using a batch of 1K uniformly sampled observations from the valid state space, identical to the batch used for success coverage. Our experiments show that DRAG consistently improves this score over baseline encoders across environments. These results will be added in the appendix.
>
>
>
> **Table 1: Mean trustworthiness of obtained representations across environments after 4M training steps**
> | Methods | trustworthiness [Venna & Kaski. 2001]
> |----------|----------|
> | **DRAG**    | **98.7%**   |
> | SKEWFIT    | 93.3%   |
> | RIG    | 88.5%   |
>
>
>
>
> ***Question 4: On ablations and hyperparameter sensitivity***
>
> We thank the reviewer for highlighting this point. Ablation studies and sensitivity analyses are indeed important to assess the contribution of individual components in our framework.
>
> We provide these analyses in the appendix:
>
> - **Figure 12** presents a sensitivity analysis over the main hyperparameter λ, which controls the strength of distributional robustness. DRAG performs best for λ values between 10 and 100, with optimal coverage at λ=100. Smaller values can lead to instability due to excessive focus on rare samples, while larger values overly regularize toward the marginal distribution p(x), limiting exploration. For comparison, SKEWFIT performs best at α=−1, corresponding to λ=1 in the DRO formulation (see Appendix C). This difference stems from SKEWFIT’s pointwise posterior estimation, which requires less aggressive reweighting than the smoother, neural-weighted distribution used by DRAG
> - **Figure 13** compares the performance of DRAG and SKEWFIT when varying the number of Monte Carlo samples M used to estimate the generative posterior. While increasing M (from 1 to 10) improves the accuracy of the likelihood estimate, it does not lead to better agent performance—in fact, it slightly reduces success coverage on average. This suggests that the robustness of DRAG is not due to more accurate pointwise estimations, but rather to the smoother weighting function induced by its neural predictor, which better generalizes across similar inputs. In contrast, SKEWFIT’s pointwise estimates can lead to abrupt weight changes, especially for poorly represented inputs.
> - **Figure 7** studies the influence of smoothing the VAE encoder used during action selection. We employ a delayed encoder updated via EMA to stabilize latent representations, using hyper-parameter τ. The best performance is obtained with τ=0.05; larger values (e.g., τ=1) make the representation unstable, while smaller values (e.g, τ=0.01) slow down adaptation excessively.
>
> We will make sure to better highlight these results in the main paper in the final version.

---

> > ### Comment · Reviewer_pMkP · 2025-08-02
> >
> > Thanks for the response! The authors have addressed all my concerns. I will increase my rating to 4.

---

> > > ### Author Response · Authors · 2025-08-07
> > >
> > > Thank you for your updated score and feedback. We're glad the clarifications were helpful and remain happy to discuss further if needed.

---

### Official Review · Reviewer_u4AE · 2025-07-02

**Clarity:** 2
**Significance:** 3
**Originality:** 3
**Rating:** 4
**Confidence:** 3

**Summary:**

This paper introduces DRAG, a method for goal-conditioned reinforcement learning that addresses exploration challenges in high-dimensional state spaces by combining a $\beta$-VAE with DRO. Unlike traditional approaches where auto-encoders overfit to frequently visited states, DRAG uses an adversarial neural weighter to prioritize underrepresented states during training, ensuring broader state-space coverage and more effective goal sampling. The method eliminates the need for explicit exploration strategies by leveraging DRO to guide latent-space learning, enabling the policy to generalize to unseen goals. Experiments in maze navigation and robotic control tasks demonstrate that DRAG outperforms baselines like RIG and Skew-Fit in both state coverage and downstream task performance, even without pre-training or environment priors.

**Questions:**

1. The paper claims DRAG improves "state space coverage and downstream control performance"—could you clarify what "downstream control" specifically refers to? Is it the policy’s ability to reach arbitrary goals sampled from the latent space? Or does it include task-specific performance (e.g., maze exit) when the policy is fine-tuned with sparse rewards?

2. The background/related work section is somewhat extensive. Have you considered moving parts of it to the appendices to improve readability? Meanwhile, Figures 8 and 9 seem critical for understanding DRAG’s contributions. Why were they placed in the appendix instead of the main text? Their visual impact could help readers grasp the method’s advantages immediately.

**Ethical Concerns:**

["NO or VERY MINOR ethics concerns only"]

**Final Justification:**

I'll keep my original score

**Limitations:**

Yes

**Paper Formatting Concerns:**

No major formatting issue.

**Quality:**

2

**Strengths And Weaknesses:**

**Strengths:**

This work advances RL by addressing the valuable yet understudied challenge of improving state coverage for VAEs in RL, which has broad implications for both model-free and model-based RL. The paper provides both theoretical derivations and experiments for adversarial reweighting, demonstrating its effectiveness in hard-exploration tasks. By ensuring the VAE’s latent space comprehensively represents the environment, DRAG unlocks more future study on reliable goal-conditioned policies.


**Weaknesses:**

1. Unclear contribution statement: The introduction’s contribution list is somewhat disorganized, making it hard to discern the core novelty (DRAG via DRO-VAE) from secondary insights (e.g., reinterpreting Skew-Fit). A clearer structure would first present DRAG as the main contribution, then contextualize it against baselines (Skew-Fit as a non-parametric DRO-VAE instance), and finally summarize empirical results.

2. The experimental validation is conducted exclusively in controlled, simplistic environments (2D mazes and single robotic arm tasks with static backgrounds), which raises significant questions about the method's generalizability to more realistic scenarios. Some limitations emerge from this narrow scope:
    - The environments use clean, synthetic visual inputs without real-world challenges like camera viewpoint changes, dynamic backgrounds, visual distractors (e.g., moving objects), etc. This leaves untested whether the adversarial weighter can maintain robust performance when the state representation must handle noisy, high-variance visual inputs common in real applications.
    - The static environments don't evaluate DRAG's ability to handle time-varying visual features, moving targets or obstacles, and non-stationary dynamics. Such conditions could particularly challenge the DRO component's assumption of a coherent uncertainty set.

---

> ### Author Rebuttal · Authors · 2025-07-30
>
> We thank the reviewer for their insightful comments and suggestions. We address their points below:
>
> **Regarding experimental validation and generalization:**
>
> We thank the reviewer for these insightful suggestions. To partially address them within the rebuttal timeframe, we performed additional experiments where input images were perturbed by noise, turning each pixel black with probability $p=0.2$ according to a Bernoulli distribution. The results show that while all methods (DRAG, SKEWFIT, DRO) experience some slight performance degradation under noise, the relative comparative trends remain unchanged, with DRAG maintaining its advantage.
>
> **Table 4: Mean Success Coverage across Noisy Maze and Robotic Environments after 4M training steps**
> | Methods | Success coverage (with noise) | Success coverage (without noise)
> | --------- | ---------------- | ---------------- |
> | **DRAG**  | **72%**          | **81%**          |
> | Skew-FIT  | 59%              | 66%              |
> | RIG       | 36%              | 42%              |
>
> Regarding distractors, moving targets, and dynamic obstacles, these are indeed challenging aspects. However, there is no clear reason why DRAG would lose its edge in such scenarios. Our method operates in a fully unsupervised setting where any observed state is a potential goal to reach, enabling preparation for a broad range of future tasks. The aim of this paper is not to propose methods for state selection or skill decomposition, but rather to maximize coverage given a set of states. Introducing uncontrollable distractors naturally expands the state and goal space, increasing the problem complexity. While interesting, this lies somewhat beyond the scope of the current study.
>
>
> Regarding the handling of time-varying visual features and non-stationary dynamics, we agree that this is an interesting direction of analysis, which can pose challenges to many RL approaches based on learned representations. Testing DRAG’s ability to recover after abrupt changes in dynamics—such as adding a wall in a maze, as done in SVGG [Castanet et al., 2023]—is a promising direction for future work to further validate robustness. However, we emphasize that the vanilla version of DRAG is not specifically designed to handle non-stationary dynamics. That said, when combined with a dedicated goal selection strategy like the SVGG criterion used in Section 4.2 of our paper (which leverages the entropy of a success predictor), there is reason to believe it could recover effectively.
>
> **On the meaning of “downstream control”:**
>
> By “downstream control,” we specifically refer to the policy’s ability to reach arbitrary goals sampled from the learned latent goal space. This captures goal-conditioned control performance (accuracy and efficiency in reaching goals). We do not include task-specific fine-tuning performance with sparse rewards in this definition. We will clarify this point in the revised manuscript (we agree that “downstream control” may be misleading).
>
> **On related work placement and appendix figures:**
>
> Figures 8 and 9 were placed in the appendix primarily due to their size (occupying a full page each). We agree these figures are important to visually convey DRAG’s advantages. As the final version allows one extra page, we will prioritize moving these figures to the main text to improve accessibility and impact.
>
> Thank you again for your constructive feedback.

---

> > ### Comment · Reviewer_u4AE · 2025-08-06
> >
> > I appreciate the authors' effort in the rebuttal. I agree that DRAG is an unsupervised method, yet the assumption of trying to cover the observation space uniformly with the latent space may not work in more generalized settings. The environments used in this work are spatially structured, so that the latent coverage can be achieved rather easily. I believe more evidence is required to make this method a general one. Thus I will maintain my score as it is.

---

> > > ### Author Response · Authors · 2025-08-07
> > >
> > > Thank you for your thoughtful comment. We would like to clarify that our main contribution is to leverage distributionally robust optimization (DRO) to promote better latent space expansion and avoid collapse during exploration in goal-conditioned RL. Our method is particularly well suited for image-based environments with spatial structure, such as robotic control and navigation, where VAEs provide a meaningful latent space. However, even in such settings, achieving good coverage is challenging. Without anticipating distribution shifts, the representation can collapse, hindering exploration — which is precisely what our DRO-based approach aims to mitigate.
> > >
> > > We agree that generalizing to more abstract or non-spatial environments would require further investigation, potentially using different encoders (e.g., contrastive or dynamics-aware methods like forward-backward models). Still, we believe the core DRO principle remains broadly applicable.
> > >
> > > We will clarify this in the final version. Thank you again for your valuable feedback.

---

### Official Review · Reviewer_xUWH · 2025-07-02

**Clarity:** 3
**Significance:** 2
**Originality:** 2
**Rating:** 5
**Confidence:** 4

**Summary:**

This work contributes DRAG, a novel method for online, intrinsically-motivated, goal-conditioned reinforcement learning from high-dimensional inputs (such as images). In particular, the method leverages the framework of distributional robust optimization (DRO), which allows supervised learning models to deal with distributional shift by optimizing it against worst-case data distributions. The authors extend this framework for online reinforcement learning tasks by integrating a VAE model, thus allowing the sampling of novel goals in the latent space during training of the RL agent. The authors evaluate DRAG on 2D mazes and a robotic tasks across two different dimensions: (i) whether DRAG outperforms other methods is reaching a wide range of goals at execution time; (ii) how do different selection strategies of latent goals improve (or not) the performance of DRAG. The results show (i) how DRAG is able to achieve a higher coverage of different goals, while being more stable than the baselines; (ii) the performance of DRAG can be improved with particular resampling strategies.

**Questions:**

I am willing to further revise my score if the authors positively address my comments in "Strengths And Weaknesses", especially focused on:
1) Can the authors clarify the scope of their contribution and its novelty?
2) How does DRAG compare against the other baseline methods in sample and time complexity?

**Ethical Concerns:**

["NO or VERY MINOR ethics concerns only"]

**Final Justification:**

The authors introduce a novel method for intrinsically-motivated goal-conditioned reinforcement learning from high-dimensional inputs. My main concerns with the paper (inconclusive novelty) were addressed by the authors. As such, I improved by score to 5.

**Limitations:**

The authors don't provide an explicit limitations section or subsection in the main document, only in appendix, which is unfortunate. As previously mentioned, I would consider discussing the computational complexity of the method against other standard methods in online goal-conditioned RL, as it provide a clearer view of pros and cons of the method for a practitioner or researcher in the field.

**Quality:**

3

**Strengths And Weaknesses:**

**Strengths**:
- Well written paper, with a (welcomed) background section on DRO;
- Interesting results, demonstrating a clear improvement over the selected baselines;
- Extensive evaluation (even if most in Appendix).

**Weaknesses**:
- Limited novelty (see comments).

**Comments**
-  Overall I thoroughly enjoyed this paper, which I found well written and without any major typos. However, I would like to point out some minor details that are currently missing (see "Typos and Minor Comments" below).
- The authors do a great job at describing all necessary background information to understand the paper, in particular the discussion on DRO (Section 2.4) which was most welcomed. However, this also leads to my biggest concern with the paper, which resides in its apparent lack of novelty: the current version of the work seems to pass the idea that the core contribution of the paper is the integration of a VAE (to process high-dimensional image observations and sample goals from the prior) in the (already existing) DRO framework. The related works of Section 2.2 and Section 2.3 also do not highlight clearly the differences between previous works and the proposed method. This is exacerbated by a confusing "contributions" summary part of the introduction, where point number 1 seems exactly the same as point number 3. My recommendation would be to have a dedicated related work section, where the authors clarify the differences between the existing methods and their proposed approach, instead of mixing background and related work. Moreover, I would streamline the "contributions" summary of the introduction.
- Despite not being evaluated on a wide range of goal-conditioned tasks, the results on the available tasks show a significant improvement over other, DRO-like, approaches. I would recommend the authors to highlight some of the main results of the experiments available in the appendix in the main text, in particular of the hyperparameter ablation studies and the latent space visualizations, which I found interesting. However, there is one evaluation that I believe is missing from the paper: what is the time complexity of the method against the baselines? Since DRAG requires Monte-Carlo sampling of latent codes for each data-point in the mini-batch, I would assume that it's a more expensive approach than the baselines. There are some results regarding the performance of DRAG with different values of $m$ (Appendix D2) but not regarding computational time.

**Typos and Minor Comments**
- What do the authors mean with "collaborative setting" (line 53)? Collaborative settings in regards to RL has a specific meaning which I don't believe is what the authors what to convey.
- (Line 234) $\mathcal{L}_{\theta, \phi, r}^{\text{DRO-VAE}}$ is not defined in any equation.
- The use of $S$ for the state space in the extended MDP formulation and as the symbol for the success coverage metric (line 267) is unfortunate.
- A lot of in-line equations are broken (line 178, 225, 226, 308, 311).
- The caption of Figure 1 is quite generic... Does it represent your DRAG framework? If so, it should be mentioned in the caption.
- In Appendix, the title of Table 3 is not in Camel-Case. Also, the structure is a bit confusing: there is an example of hyperparameter ablation (Figure 7) in Appendix A.2.2. and then a whole section dedicated to hyperparameter ablations in Section D.2. Perhaps, it can be optimized.

---

> ### Author Rebuttal · Authors · 2025-07-30
>
> We thank the reviewer for the thorough and constructive feedback. We address the main concerns below.
>
> **1. On Novelty and Related Work Presentation**
>
> We thank the reviewer for raising concerns about the novelty and the structure of our related work discussion. While it is true that our method builds upon established components (β-VAEs, DRO principles, and goal-conditioned RL agents), we believe our contribution is novel in both formulation and application.
>
> Our key innovation lies in the integration of DRO into the online training of VAEs within reinforcement learning, which—to the best of our knowledge—has not been explored before, even outside of RL. Applying DRO in this context is non-trivial, as it requires careful adaptation of the min-max objective to variational training with approximate likelihoods, as well as efficient sampling and optimization strategies in the presence of non-stationary replay buffers. This results in a principled alternative to prior  approaches, such as SKEWFIT, which we show to be a particular instance of our more general framework.
>
> We also agree that the distinctions between our approach and related methods could be highlighted more clearly in Sections 2.2 and 2.3. We will revise these sections as follows:
>
> - Section 2.2, which focuses on intrinsically motivated agents and goal resampling strategies, will explicitly point out that these methods are difficult to scale to high-dimensional observations such as images, due to the curse of dimensionality. A natural idea is to perform such resampling in a compressed latent space. However, we show that this is far from trivial when the representation is learned online, as it can collapse or become biased, especially when exploration induces significant distribution shifts. This motivates the need for principled control over representation learning during training.
> - Section 2.3, on online representation learning with VAEs, positions our method in the landscape of prior VAE-based methods like SKEWFIT. We will emphasize that naively training the encoder via the ELBO objective fails to account for the non-stationarity of the data, and leads to representations that underfit the full diversity of visited states. SKEWFIT proposes to mitigate this via a handcrafted reweighting scheme designed to approximate a uniform goal distribution. However, as we show, this corresponds to a particular instantiation of a more general DRO objective, which can lead to unstable training. One of our contributions is to formalize this connection and leverage the theoretical tools of DRO to derive a more stable and general reweighting strategy, grounded in optimization theory.
>
> Regarding the clarity of the stated contributions, we acknowledge that Contribution 1 (DRO formulation for VAEs) and Contribution 3 (practical algorithm) may appear overlapping. We will revise the introduction to clarify the distinction:
>
> - Contribution 1 provides the theoretical formulation that adapts DRO principles to VAE-based representation learning, establishing a new optimization objective that subsumes prior methods.
> - Contribution 3 describes the algorithmic realization of this framework in the DRAG agent, which combines VAE-based goal encoding, neural importance weighting, and online training from replay buffers.
>
> Finally, we take note of the suggestion to better separate background and related work. While our current structure aimed to streamline exposition, we agree that a clearer separation (e.g., explicitly distinguishing background from prior art, or deferring part of it to an appendix) may improve readability and will consider this in the camera-ready version.
>
>
> **2. Experimental Scope and Ablations in Main Text**
>
> We agree that highlighting key results from the appendix in the main paper would improve clarity. We will use the extra page to incorporate in the main text:
> - A short discussion of latent space visualizations.
> - A summary of the ablation study findings (e.g., optimal values for $\lambda$, stability across $\beta$, the number of samples $M$ per estimation, etc.).
>
> **3. Runtime and Computational Cost**
>
> We agree with the reviewer that we should also include runtime measurement, to assess the impact of the overhead induced by the additional likelihood estimation (for SKEWFIT and DRAG compared to RIG) and the use of a neural weighter (for DRAG). With additional experiments reported in table 1 below (values are averaged over 6 seeds on each environment), we observe  that DRAG and SKEWFIT require on average about 200 seconds for 20k steps, against 176 seconds with RIG. This slight overhead is negligible compared to the time spent in environment interactions. This is because we  only use one sample (i.e. $M=1$) for the likelihood estimation in DRAG and SKEWFIT (please refer to the experiments with $M=10$ in Appendix D.2.2, which shows that using more samples does not impact the results of both methods). Also, no notable difference in runtime is observed when comparing DRAG to SKEWFIT.
>
> **Table 1 : Mean execution runtime on V100 GPU (32GB) for MAZE and METAWORLD environments**
> | Method          | Time per 20K steps (s)  | Time for 4M steps (s) |
> |-----------------|------------------------|------------------------|
> | DRAG  | 211                    | 42,200  (11.72 h)                   |
> | SKEWFIT  | 208                    | 41,600  (11.55 h)                   |
> | RIG             | 176                    | 35,200  (9.77 h)                   |
>
>
>
> To better link performance to runtime, Table 2 shows success coverage over wall-clock time (same runs as Fig. 3):
>
> **Table 2 : Success coverage / runtime for MAZE and METAWORLD environments**
> | Hour | **DRAG** (%) | Skew-FIT (%) | RIG (%) |
> |------|----------|--------------|---------|
> | 1    | **34**       | 31           | 26      |
> | 3    | **45**       | 42           | 28      |
> | 5    | **60**       | 50           | 32      |
> | 7    | **70**       | 58           | 36      |
> | 9    | **77**       | 63           | 40      |
> | 11   | **81**       | 66           | 42      |
>
>
> We will add this additional analysis in Appendix D.
>
> **4. Clarification on “Collaborative Setting”**
>
> We thank the reviewer for pointing this out. Our intention was to emphasize the mutual interaction between the representation learner and the RL agent. However, we acknowledge that “collaborative” may be misleading in the RL context, where it often implies multi-agent cooperation. We will rephrase the sentence to avoid confusion.
>
> **5. Definition of $L_{\theta, \phi, r}^{\text{DRO-VAE}}$**
>
> $L_{\theta, \phi, r}^{\text{DRO-VAE}}$ refers to the quantity introduced in Equation (5). We agree that the phrasing at line 234 could be clearer and will revise it to:
>
> “…where this approximated lower bound, which we call hereafter $L_{\theta, \phi, r}^{\text{DRO-VAE}}({x_i}_{i=1}^n)$, can be estimated at each step via Monte Carlo…”
>
> **6. Notation and Formatting Issues**
>
> - We agree that using $S$ both for state space and success coverage metric is unfortunate. We will change the success metric notation (e.g., to $C$) for clarity.
> - Broken in-line equations (lines 178, 225, 226, 308, 311): thank you, we will fix these in the final version.
> - Figure 1 caption: we confirm that it illustrates the DRAG framework, as it includes "DRO sampling" in the red part. We will clarify this.
> - Appendix Table 3 title and section organization: thanks for pointing this out. We will improve the formatting and merge/restructure ablation-related sections for clarity.

---

> > ### Comment · Reviewer_xUWH · 2025-08-04
> > **Acknowledgment of Author's Rebuttal**
> >
> > Thank you for addressing my comments and for the extra ablation studies performed. As the reviewers have successfully addressed my comments, I raise the score to 5.

---

> > > ### Author Response · Authors · 2025-08-07
> > >
> > > Thank you for your updated score and feedback. We're glad the clarifications were helpful and remain happy to discuss further if needed.

---

> ### Author Response · Authors · 2025-08-08
>
> Thank you again for your kind and encouraging comments.
> However, as far as we can see from the author interface, the score has not been updated as you mentioned in your previous comment.
> If you still intend to raise it, we’d be very grateful if you could kindly submit the change when you get a chance. Many thanks in advance.

---

### Official Review · Reviewer_13ri · 2025-07-03

**Clarity:** 3
**Significance:** 3
**Originality:** 3
**Rating:** 4
**Confidence:** 3

**Summary:**

The paper introduces DRAG, an online goal-conditioned RL method that couples a beta-VAE with distributionally robust optimisation (DRO). A small neural weighter assigns adversarial importance weights to replay-buffer images so that VAE training emphasizes under-represented states, effectively steering the latent space toward a uniform coverage of the environment. The authors derive a DRO-VAE objective and show that SKEW-FIT is its non-parametric special case, propose the parametric DRAG variant for greater stability and integrate DRAG into a RIG-like GCRL loop with latent-goal sampling. They empirically demonstrate markedly higher success-coverage and improved downstream control on four pixel PointMazes and a Reach-Hard-Walls robot task, compared with RIG and SKEW-FIT.

**Questions:**

- How sensitive is DRAG to the temperature of the KL relaxation and to the beta of the VAE? A small study varying these would clarify robustness.
- What is the wall-clock overhead of training the neural weighter versus plain RIG? Please report GPU hours or frames per second.

**Ethical Concerns:**

["NO or VERY MINOR ethics concerns only"]

**Final Justification:**

After rebuttal, my main concerns are addressed. The methods remains technically sound but the evaluation is still limited in scope. I keep 4.

**Limitations:**

yes

**Quality:**

3

**Strengths And Weaknesses:**

Quality:
- Strengths: sound formulation with DRO-VAE derivation following standard DRO theory, experiments run six seeds and report confidence intervals with RLiable statistics.
- Weaknesses: limited scope with small-scale visual tasks (82 × 82 images) and a single robotic manipulation variant, no runtime or sample-efficiency analysis is provided, computational overhead of the weighter network is not quantified.

Clarity:
- Strengths: well organised paper, the relation between SKEW-FIT and DRAG is explicitly proved, figures 3-4 summarize results effectively.
- Weakness: a clear presentation of the exact ELBO and hyperparamters used would help reproducibility

Significance:
- Strengths: tackling exploration bottlenecks in online visual GCRL without separate explorers is an important practical problem, the idea of casting state-coverage as DRO over VAE training data is novel in this context and could inspire future work in representation learning for RL.
- Weakness: impact is moderated by the modest experimental breadth and by missing comparisons with other contemporary representation learners

Originality:
- Strengths: first application of parametric DRO to auto-encoder training in RL, the analytical link to SKEW-FIT clarifies prior work.
- Weakness: the individual ingredients (beta-VAE, DRO and RIG) are established techniques and novelty lies mainly in their combination.

---

> ### Author Rebuttal · Authors · 2025-07-30
>
> Many thanks for your thorough and constructive feedback. We address the main concerns below.
>
> **Weakness: Quality (and Question about runtime overhead)**
>
> - ***Limited scope (image resolution, task diversity):***
>
> Thanks for the suggestion for considering higher resolution images to assess the ability of the method to deal with larger inputs. To complement our experiments, we ran the three methods (i.e. RIG, SKEWFIT and DRAG) on all environments with a higher image resolution (128x128 instead of 82x82 in the paper). We obtained the following results, averaged on every environment from the paper:
>
> | Methods | 1M  | 2M | 3M |4M steps
> |----------|----------|----------|-------|----------|
> | **DRAG**    | **42%**   | **58%** | **71%**  | **79%**
> | **↳ 82x82 comparison**    | **46%**   | **66%** | **74%**  | **81%**
> | SKEWFIT    | 28%   | 44%  |57% | 65%
> | ↳ 82x82 comparison    | 33%   | 51%  |59% | 66%
> | RIG    | 22%   | 29%   |35%| 38%
> |  ↳ 82x82 comparison    | 28%   | 34%   |39%| 42%
>
> **Table 1: Mean success coverage across maze and robotic environment for 128x128 pixels observations (success coverage for 82x82 is given for comparison)**
>
>
> All methods degrade slightly with higher resolution, but DRAG retains a clear advantage.
>
> We agree that broader task diversity would further validate generality. While our study focuses on goal-conditioned robotic tasks, the DRO-based representation principle is generic and could be extended to navigation or multi-agent setups, which we leave for future work.
>
>
> - ***Sample-efficiency and runtime analysis***
>
> First, we would like to respectfully highlight that Figures 3 and 4 already inform about sample efficiency as the x-axis shows the number of steps (or environment interactions). Our approach appears significantly more sample efficient than baselines.
>
> However, we agree with the reviewer that we should also include runtime measurement, to assess the impact of the overhead induced by the additional likelihood estimation (for SKEWFIT and DRAG compared to RIG) and the use of a neural weighter (for DRAG). With additional experiments reported in Table 2 below (values are averaged over 6 seeds on each environment), we observe  that DRAG and SKEWFIT require on average about 200 seconds for 20k steps, against 176 seconds for RIG. This slight overhead is negligible compared to the time spent in environment interactions. This is because we only use one sample (i.e. $M=1$) for the likelihood estimation in DRAG and SKEWFIT (please refer to the experiments with $M=10$ in Appendix D.2.2, which shows that using more samples does not impact the results of both methods). Also, no notable difference in runtime is observed when comparing DRAG to SKEWFIT.
>
> **Table 2 : Mean execution runtime on V100 GPU (32GB) for MAZE and METAWORLD environments**
> | Method          | Time per 20K steps (s)  | Time for 4M steps (s) |
> |-----------------|------------------------|------------------------|
> | DRAG  | 211                    | 42,200  (11.72 h)                   |
> | SKEWFIT  | 208                    | 41,600  (11.55 h)                   |
> | RIG             | 176                    | 35,200  (9.77 h)                   |
>
>
>
> To better link performance to runtime, Table 3 shows success coverage over wall-clock time (82×82, same runs as Fig. 3):
>
> **Table 3 : Success coverage / runtime for MAZE and METAWORLD environments**
> | Hour | **DRAG** (%) | SKEWFIT (%) | RIG (%) |
> |------|----------|--------------|---------|
> | 1    | **34**       | 31           | 26      |
> | 3    | **45**       | 42           | 28      |
> | 5    | **60**       | 50           | 32      |
> | 7    | **70**       | 58           | 36      |
> | 9    | **77**       | 63           | 40      |
> | 11   | **81**       | 66           | 42      |
>
>
> We will add this additional analysis in Appendix D.
>
>
> **Weakness: Clarity on ELBO and hyperparameters:** "a clear presentation of the exact ELBO and hyperparamters used would help reproducibility"
>
> We appreciate the request for more clarity, but we are not sure to fully understand what the reviewer means when asking for the exact ELBO. The ELBO used for VAE training is given in Eq. (5), which is the standard form for a VAE, but weighted by the learned reweighting function $r(x)$.
>
> - ***Method Derivation***
>
> If the reviewer wants more explanation about the derivation of the method, we can complement the paper starting from Eq. (3), that we relax with a KL term as in (1), to get:
> $\min_{\theta, \phi} \max_{r:E_{p}r = 1} - E_{x \sim p} \ r(x) \log{p_{\theta,\phi}(x)} - \lambda E_{x \sim p} \ r(x) \log r(x)$
>
> $=\max_{\theta, \phi} \min_{r:E_{p}r = 1}  E_{x \sim p} \ r(x) \log{p_{\theta,\phi}(x)} + \lambda E_{x \sim p} \ r(x) \log r(x)$
>
> $=\max_{\theta, \phi} \min_{r:E_{p}r = 1}  L_{\theta,\phi,r}$
>
> where $L_{\theta,\phi,r}=E_{x \sim p} \ r(x) \log{p_{\theta,\phi}(x)} + \lambda E_{x \sim p} \ r(x) \log r(x)$
>
> Then, we could write a lower bound as:
> $L_{\theta,\phi,r} \geq E_{x \sim p(x)} r(x) E_{z \sim q_\phi(z|x)} \log \frac{p(z) p_\theta(x|z)}{q_\phi(z|x)} +  \lambda E_{x \sim p} \ r(x) \log r(x)$
>
> However, using this quantity to optimize the weighter $r$ would not be valid, as minimizing a lower bound makes no sense. This led us to an alternate learning scheme that optimizes $r$ on an approximation of the true likelihood of any sample $x$ (which we construct with M samples of code $z$ for each $x$), as defined in Eq. (4), while the weighted VAE is optimized on the weighted ELBO given in Eq. (5) for a fixed weighter.
>
> - ***Reproducibility (Algorithm and Hyper-parameters)***
>
> Maybe the reviewer is more interested by the full algorithm that we employ in our experiments (as the global weakness refers to reproducibility). Please refer to **Appendix B for the full pseudo-code of our method**, and to **Appendix A.3 for all hyper-parameters** used in our method and the baselines considered in our experiments.
>
>
> **Weakness: Originality**
>
> While we build upon existing components (β-VAE, DRO, RIG), our key contribution is the novel application of DRO to VAE training in online RL, which—unlike classification—presents unique challenges due to dynamic data distributions and non-stationarity (induced by the progress of the agent). This connection is non-trivial, and to the best of our knowledge, has not been explored before.
>
> Our theoretical analysis also highlights how SKEWFIT can be seen as a particular case of DRO under specific choices of $r$, allowing us to generalize and improve it using the DRO framework. The use of DRO in the setting of a VAE is also a contribution, for which we proposed a specific learning scheme, alternating the optimization of the weighter on an estimate of the true likelihood a depicted in Eq. (4) (since minimizing a lower bound for $r$ would make no sense) and optimizing the encoder-decoder on a weighted ELBO (Eq. (5)).
>
> **Weakness Significance: Comparison with other representation learning paradigms**
>
> We agree that comparing against other contemporary representation learning approaches could enrich the analysis. Notably, contrastive methods [Stooke et al., 2021] and Forward-Backward approaches [Touati and Ollivier, 2021] aim to incorporate dynamics by bringing temporally adjacent states closer or by modeling universal rewards. However, these methods generally assume access to transitions from a representative part of the environment. To our knowledge, they do not include any explicit mechanism to avoid collapse onto narrow parts of the state space—a critical issue in hard exploration settings without expert priors.
>
> Addressing this limitation is precisely the aim of our DRO-based reweighting, which promotes state space coverage even in sparse reward regimes. While our implementation focuses on β-VAEs for interpretability and disentanglement, our DRO framework is general and not tied to a specific representation architecture.
>
> In fact, any representation learning method trained from replay buffer samples using a loss $\ell(f_\theta(x))$ can be reweighted using the DRO objective:
>
> $\min_{\theta} \max_{r} \frac{1}{N} \sum_{i=1}^N r(x_i) (\ell(f_\theta(x_i))) - \lambda  r(x_i) \log r(x_i) \qquad \text{s.t.   }
> \frac{1}{N} \sum_{i=1}^N r(x_i) = 1,$
>
> This includes diffusion-based decoders, contrastive losses (e.g., InfoNCE), and temporal-difference-based objectives (e.g., in Forward-Backward RL). For generative losses, as in VAEs or diffusion models, the optimization must rely on likelihood estimates (i.e., $\log p_{\theta,\phi}(x)$), similarly to our alternate optimization strategy described in Eq. (4).
>
> Extending our DRO approach to contrastive or dynamic-aware representation learning is a promising direction we aim to pursue in future work.
>
> [Touati and Ollivier, 2021] Touati, Ahmed, and Ollivier, Yann. "Learning one representation to optimize all rewards." Advances in Neural Information Processing Systems 34 (2021): 13-23.
>
> [Stooke et al., 2021] Stooke, Adam, et al. "Decoupling representation learning from reinforcement learning." International conference on machine learning. PMLR, 2021.
>
> **Question – Sensitivity to hyperparameters (KL temperature, β)**
>
> We report a detailed sensitivity analysis over the DRO parameter λ in Appendix D.2.1. We show that values between 10 and 100 yield the most consistent results across environments. We will highlight this better in the main paper.
>
> For β (VAE KL weight), we performed ablations over the range β ∈ [0.5, 3]. While β = 2 yields the best performance on average across DRAG, SKEWFIT and RIG, we observed that the relative performance ranking remains unchanged across this range. We will clarify this in Appendix D.
>
> Additional ablations in the supplementary material also explore sensitivity to $M$ (the number of latent samples per input) and $\tau$ (the EMA coefficient for the delayed encoder).

---

> > ### Author Response · Authors · 2025-08-08
> >
> > Dear reviewer,
> >
> > As the author–reviewer discussion period is nearing its end, we just wanted to mention that if you have any questions about the paper or our rebuttal, we’d be very happy to address them.

---

### Decision · Program_Chairs · 2025-09-17

**Decision:**

Accept (poster)

**Comment:**

This paper integrates distributionally robust optimization into VAE-based representation learning to encourage more uniform visitation of states and improve exploration in online RL settings with complex "semantically sparse" visual input. The proposed method is motivated by the inherent nature of distribution shifts within online settings. This method highlights and addresses an important problem of representation collapse in online RL, and the baselines are appropriate and the results convincing. There were concerns about novelty -- being a combination of well-known methods -- but the authors successfully convinced the reviewers otherwise. Additional experiments wrt wall-clock time were provided and also seemed to satisfy reviewers.

I therefore recommend the paper is accepted as a poster.